# Residual-Guided Multi-Resolution Refinement of Foundation Models: A Case Study in Drought Forecasting

Wentao Gao[1]   Jiuyong Li[1]   Lin Liu[1]   Thuc Duy Le[1]   Jixue Liu[1]   Yanchang Zhao[2]   Yun Chen[3]

## Abstract

Regional climate prediction presents unique challenges for time series foundation models, which typically process temporal patterns through single-pass inference. Expert climatologists, in contrast, employ multi-scale temporal analysis and iterative refinement based on systematic error diagnosis. We present RGMR (Residual-Guided Multi-Resolution Refinement), an inference-time framework that adapts pre-trained foundation models to perform structured coarse-to-fine refinement for climate forecasting without updating backbone parameters. Applied to drought forecasting using the Standardized Precipitation Evapotranspiration Index (SPEI), RGMR is architecture-agnostic across the three TSFM backbones evaluated per site (TimesFM, TimeGPT, TabPFN) and consistently lowers test-set MSE on three South Australian sites and three additional regions outside South Australia. Applied to TimesFM, the wrapper reduces one-month-ahead SPEI MSE by up to 18.9% across the three South Australian sites (mean reduction ≈18.7%). Overall, RGMR provides a practical route for deploying frozen TSFMs in regional climate forecasting workflows.

## 1. Introduction

Climate prediction typically involves multi-stage analysis where meteorologists examine broad atmospheric patterns before progressively focusing on regional details, continuously adjusting forecasts as new evidence emerges (Lynch, 2006; Bauer et al., 2015). This iterative approach, where predictions evolve through structured analysis at multiple temporal scales, differs substantially from current AI

[1]Adelaide University, Adelaide, SA, Australia [2]CSIRO Technology, Canberra, Australia [3]CSIRO Environment, Canberra, Australia. Correspondence to: Wentao Gao <wentao.gao@adelaide.edu.au>.

*Proceedings of the 43rd International Conference on Machine Learning*, Seoul, South Korea. PMLR 306, 2026. Copyright 2026 by the author(s).

systems that generate forecasts through single forward passes (Schultz et al., 2021).

Recent advances in time series foundation models (TSFMs) have demonstrated strong performance across diverse forecasting domains (Das et al., 2024; Rasul et al., 2024; Ansari et al., 2024). However, TSFMs face challenges in regional climate prediction tasks such as drought forecasting, applications requiring nuanced understanding of local climate dynamics and multi-scale temporal patterns (Vicente-Serrano et al., 2020; Mukherjee et al., 2018; Gao et al., 2022; Chen et al., 2025).

We hypothesize that this limitation stems from architectural differences in how these models process temporal information. Foundation models typically map historical sequences to future predictions in a single forward pass, processing all temporal information uniformly (Lim et al., 2021). Climatological analysis, by contrast, often involves iterative examination of data at multiple time scales (Trenberth et al., 2014), identification of systematic patterns and biases (Jolliffe & Stephenson, 2003), and progressive refinement through focused analysis of specific temporal components (Palmer, 1998).

Recent work in iterative refinement for sequential prediction tasks (Wei et al., 2022; Kojima et al., 2022; Yao et al., 2023) suggests that multi-stage processing can improve performance on complex reasoning tasks. This motivates our central question: *Can time series foundation models benefit from hierarchical refinement for climate forecasting?*

Figure 1 illustrates the difference between multi-scale temporal analysis and single-pass processing using SPEI time series data. Climatological analysis typically decomposes multi-scale climate patterns into constituent temporal components (annual cycles, ENSO oscillations, and extreme events), enabling systematic examination of each scale before integration. Current AI models process the entire complex signal simultaneously, which may lead to systematic prediction errors during drought events and regime shifts.

Together, these observations motivate an error-corrective inference loop for frozen TSFMs. Iterative refinement in language tasks suggests that initial hypotheses can be evaluated and sharpened through feedback (Shinn et al., 2023;

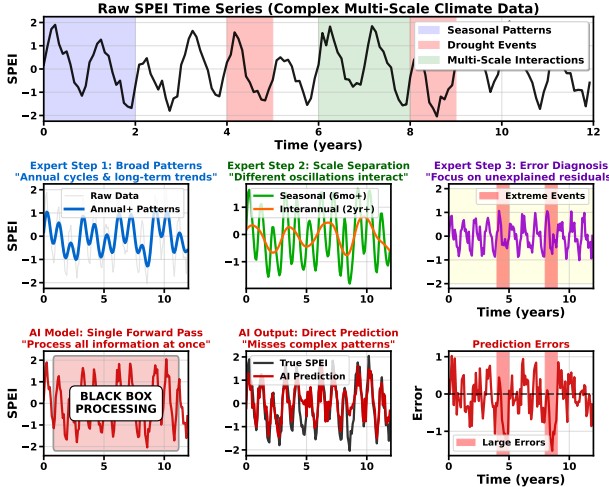

*Figure 1.* Comparison of temporal processing approaches in climate forecasting. Multi-scale decomposition analysis (middle row) versus single forward pass processing (bottom row) for complex climate pattern prediction.

Madaan et al., 2023), while operational climate systems require efficient inference that avoids gradient updates to the foundation backbone (Bauer et al., 2015). To address this setting, we propose **Residual-Guided Multi-Resolution Refinement (RGMR)**, a framework that enables time series foundation models to perform multi-scale corrective inference without modifying backbone parameters. Our approach uses domain knowledge to define relevant temporal scales, then iteratively refines predictions from coarse to fine resolutions through residual-guided error correction.

We evaluate RGMR on drought forecasting using SPEI (Vicente-Serrano et al., 2010) data across multiple South Australian locations. We find consistent improvements over direct foundation model application, achieving up to 18.9% error reduction in mean squared error.

**Our main contributions include:**

- We propose **RGMR**, an inference-time wrapper that adapts frozen TSFMs to regional climate forecasting through residual-guided coarse-to-fine refinement without updating backbone parameters.

- We provide a contraction-style analysis and a finest-resolution no-harm condition, together with projection and resolution-scale sensitivity analyses that justify the coarse-to-fine design.

- We show that RGMR consistently improves the evaluated frozen TSFM backbones, including up to 18.9% lower one-month-ahead SPEI MSE across the three South Australian sites, with modest inference overhead.

## 2. Related Work

**Drought forecasting.** Classical statistical models (e.g., ARIMA, SVMs) have long been used for drought indices (Nandgude et al., 2023) but depend on bespoke feature engineering and struggle with long-range dependencies. Deep architectures, including LSTMs (Hochreiter & Schmidhuber, 1997), Transformer-based forecasters (Vaswani et al., 2017; Zhou et al., 2021), and recent spatiotemporal models (Tan et al., 2025), improve expressivity but typically require task-specific training or fine-tuning to a region and target index. In contrast, we study an *inference-time* framework that adapts a frozen base model to drought forecasting without updating weights.

**Time-series foundation models (TSFMs).** General-purpose TSFMs such as TimesFM (Das et al., 2024), TimeGPT-1 (Garza et al., 2023), Lag-Llama (Rasul et al., 2024), Chronos (Ansari et al., 2024), and Timer (Liu et al., 2024c) aim for broad-domain forecasting with a single pre-trained backbone. Domain-specialized weather systems like GraphCast (Lam et al., 2023) and Pangu-Weather (Bi et al., 2023) achieve strong numerical weather prediction but are not designed as drop-in forecasters for regional drought indices and usually involve re-training or domain-specific pipelines. We target the complementary setting of *using* a general TSFM as-is and improving its outputs at inference time.

**Multi-scale and residual learning.** Multi-scale techniques capture temporal dynamics at different resolutions, ranging from Scaleformer's coarse-to-fine pathway (Shabani et al., 2023) and TimeMixer's decomposable multiscale mixing (Wang et al., 2024) to N-BEATS' hierarchical residual stacks (Oreshkin et al., 2020) and Minusformer's progressive residual refinement (Liang et al., 2024). In particular, TimeMixer decomposes multiscale series and mixes seasonal and trend components across fine-to-coarse and coarse-to-fine directions, showing the importance of explicitly exploiting complementary temporal resolutions. However, these methods typically realize multi-resolution behavior through dedicated architectures or training procedures. Our approach differs in *where* multi-resolution acts: rather than redesigning or retraining the forecasting model, we implement coarse-to-fine *refinement at inference time* on top of a frozen backbone, using residuals learned from short-window corrections to adjust long-window proposals.

**Inference-time adaptation.** Test-time adaptation (Sun et al., 2020) and lightweight prompting/tuning (Brown et al., 2020) enable on-the-fly specialization while retaining a pre-trained model's generality. For time series, few-shot or episodic adaptation has been explored mostly via fine-tuning (Oreshkin et al., 2021). Our RGMR framework performs *inference-time* adaptation: it composes multi-resolution proposals and residual corrections with adaptive residual

weighting, preserving the pre-trained TSFM and avoiding any weight updates.

## 3. Problem Definition

We study the task of forecasting the Standardized Precipitation–Evapotranspiration Index (SPEI), a widely used drought index that integrates precipitation and potential evapotranspiration into a single standardized time series. Let $\{y_t\}_{t=1}^T$ denote the monthly SPEI values at a specific region. We let the multivariate input vector $\mathbf{X}_t \in \mathbb{R}^D$ have its first channel set to the historical SPEI value $y_t$ and the remaining $D-1$ channels carry climate covariates, so the model conditions on past targets and past covariates jointly.

**Multi-step forecasting.** We consider forecasting the next $H$ steps. At forecast origin $t$, the goal is to predict the next $H$ SPEI values $\mathbf{y}_{t+1:t+H} \in \mathbb{R}^H$. A frozen foundation model $f_\theta$ maps a length-$W$ context window to an $H$-step *base forecast*:

$$\widehat{\mathbf{y}}_{t+1:t+H}^{\text{base}} = f_\theta(\mathbf{X}_{t-W+1:t}), \qquad f_\theta : \mathbb{R}^{W \times D} \to \mathbb{R}^H,$$

where $\mathbf{X}_{t-W+1:t} \in \mathbb{R}^{W \times D}$ denotes the multivariate input context (with past SPEI as its first channel; see above). The one-step case is recovered by setting $H = 1$. RGMR (Section 4) refines this base forecast into a final prediction $\widehat{\mathbf{y}}_{t+1:t+H}^{(K)}$ through coarse-to-fine residual correction; throughout the paper, $\widehat{\mathbf{y}}^{(k)}$ refers exclusively to RGMR's level-$k$ refined output. The full notation used throughout the paper is summarized in Appendix A.1 (Table 6).

**Training-time vs. inference-time residuals.** RGMR strictly separates residuals used for offline calibration from residuals available at test time. (i) The *training residual* $\widetilde{\mathbf{R}}^{(k)} = \mathbf{y} - \widetilde{\mathbf{y}}^{(k)}$ uses observed targets before test evaluation, where $\widetilde{\mathbf{y}}^{(k)}$ is the level-$k$ short-window proposal defined in Section 4.3; it is used only to fit and validate the Ridge residual predictor. (ii) The *predicted residual* $\widehat{\mathbf{R}}^{(k)} = g_\phi^{(k)}(\mathbf{z}_t^{(k)})$ is the only residual quantity available at inference, where $g_\phi^{(k)}$ and its feature vector $\mathbf{z}_t^{(k)}$ are specified in Section 4.3. Test targets $\mathbf{y}_{t+1:t+H}$ are never read when forming the current correction.

## 4. Residual-Guided Multi-Resolution Refinement (RGMR)

### 4.1. Method Overview

RGMR is an inference-time framework that adapts *frozen* time-series foundation models through hierarchical residual refinement; Figure 2 summarizes the two-stage workflow. We fix a resolution scale $\mathcal{R} = \{r_1 = 12, r_2 = 6, r_3 = 3, r_4 = 2, r_5 = 1\}$ months from coarse to fine. The five strides correspond to roughly annual ($r_1{=}12$), semi-annual ($r_2{=}6$), seasonal/sub-seasonal ($r_3{=}3$), bimonthly ($r_4{=}2$), and the native monthly resolution ($r_5{=}1$). These match the dominant temporal scales used by climatologists for SPEI analysis (Vicente-Serrano et al., 2010). For each $r_k$, the frozen backbone produces a full-context proposal. During offline calibration, we fit lightweight per-level residual predictors on short-window inputs to capture systematic, level-specific errors. At test time, no ground truth is used and no backbone weights are updated. Proposals are refined from coarse to fine by applying the learned residual predictors to the long-window proposals, with clipped adaptive weights and a fixed step-size rule. The following subsections detail the projection operator (4.2), residual learning and features (4.3), the inference-time update rule (4.4), and the contraction analysis (4.5).

### 4.2. Multi-Resolution Projection Operations

For each resolution level $r \in \mathcal{R}$, we apply temporal projection operations

$$\mathcal{P}_r = \mathcal{U}_r \circ \mathcal{D}_r, \tag{1}$$

where $\mathcal{D}_r$ downsamples by averaging within non-overlapping length-$r$ segments and $\mathcal{U}_r$ upsamples by repeating values. Using zero-based indexing (so $ir = i \times r$) and letting $\lfloor \cdot \rfloor$ denote the floor operator, we have $\mathcal{D}_r(x)[i] = \frac{1}{r} \sum_{j=ir}^{(i+1)r-1} x[j]$ and $\mathcal{U}_r(\mathcal{D}_r(x))[j] = \mathcal{D}_r(x)[\lfloor j/r \rfloor]$ for complete blocks. If the context length is not divisible by $r$, the final partial block is averaged over its available entries and repeated only up to the original sequence length. Thus $\mathcal{P}_r(\mathbf{X}_{t-W+1:t})$ always has length $W$, and no future padding is introduced. This yields multi-resolution temporal views of the input while preserving cross-channel temporal alignment and remaining $O(W)$ per level.

### 4.3. Residual Pattern Learning

For each forecast origin $t$ in the offline calibration period and each resolution level $k$, we query the frozen backbone with a *short* context to expose level-specific systematic errors:

$$\widetilde{\mathbf{y}}_{t+1:t+H}^{(k)} = f_\theta(\mathcal{P}_{r_k}(\mathbf{X}_{t-L_{\text{short}}+1:t})), \tag{2}$$

and form the training residual $\widetilde{\mathbf{R}}_{t+1:t+H}^{(k)} = \mathbf{y}_{t+1:t+H} - \widetilde{\mathbf{y}}_{t+1:t+H}^{(k)}$. We set $L_{\text{short}}{=}12$ months, one annual cycle, so that the short-window proposal misses the multi-year climate context the long-window backbone exploits and therefore exhibits the structured biases (phase, drift, regime) that the residual predictor is asked to recognize. This value is fixed across all sites and is not tuned on test data.

For each level $k$ we then fit a closed-form Ridge predictor (Hoerl & Kennard, 1970) $g_\phi^{(k)}(\mathbf{z}_t^{(k)}) = \mathbf{\Theta}^{(k)\top} \mathbf{z}_t^{(k)}$ that maps a feature vector $\mathbf{z}_t^{(k)} \in \mathbb{R}^{d_z}$ to the $H$-step training residual,

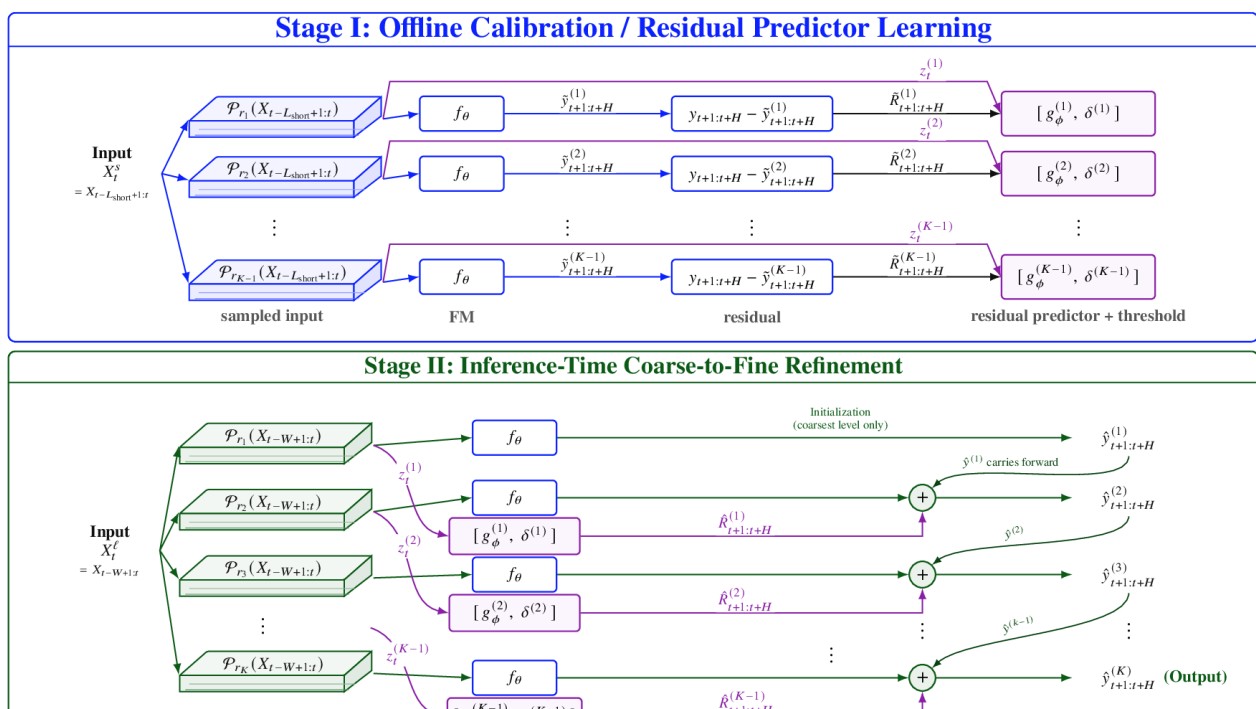

*Figure 2.* Two-stage workflow of RGMR. **Stage I:** Offline calibration learns level-wise residual correction modules from short-window projected inputs. At each resolution level, the frozen foundation model produces a short-window base forecast, and the observed forecast error is used to train a lightweight residual correction module. Feature vectors and training residuals are used to fit the residual predictors and select their validation thresholds. **Stage II:** At inference time, projected long-context inputs are processed by the frozen foundation model and the fixed residual correction modules. RGMR starts from the coarsest forecast and performs boosting-style coarse-to-fine refinement: at each refinement step, the predicted residual guides the additive correction of the forecast carried from the previous level, producing the final output. Boxes indicate FM, operators or learned models; unboxed quantities represent variables.

where $d_z$ is the feature dimension and $\mathbf{\Theta}^{(k)} \in \mathbb{R}^{d_z \times H}$ is the level-$k$ Ridge coefficient matrix,

$$\mathbf{\Theta}^{(k)} = \arg \min_{\mathbf{\Theta} \in \mathbb{R}^{d_z \times H}} \sum_{t \in \text{Train}} \left\| \widetilde{\mathbf{R}}^{(k)}_{t+1:t+H} - \mathbf{\Theta}^\top \mathbf{z}^{(k)}_t \right\|_2^2 + \lambda^{(k)} \|\mathbf{\Theta}\|_F^2.$$

For each $\lambda^{(k)}$, Ridge coefficients are fitted on the training split and selected by time-ordered validation (Bergmeir et al., 2018) from the log-grid $\{10^{-4}, 10^{-3}, \dots, 10^2\}$. Separately, we select a level-wise residual-magnitude threshold $\delta^{(k)}$ on the validation split, defined as a quantile of validation predicted-residual magnitudes and used as the soft-threshold midpoint in the adaptive residual-weighting gate in Eq. (4). Using the selected $\lambda^{(k)}$, the final residual predictor $g^{(k)}_\phi$ is refit on the union of the training and validation splits; the pair $(g^{(k)}_\phi, \delta^{(k)})$ is then frozen for test-time inference. No test targets are used for fitting or selection. The feature vector $\mathbf{z}^{(k)}_t$ (written $\mathbf{z}_t$ when the level is clear) concatenates: (i) the most recent $p_{\text{lag}}$ targets $\mathbf{y}_{t-p_{\text{lag}}+1:t}$; (ii) rolling mean and standard deviation over the last 12 months; (iii) a *linear trend coefficient* computed as the ordi-

nary least-squares slope of $\{y_{t-q+1}, \dots, y_t\}$ regressed on $\{1, \dots, q\}$ (we use $q{=}12$), capturing local low-frequency drift not represented by the rolling mean; (iv) the most recent $p_{\text{lag}}$ inference residuals available at the rolling origin (revealed in chronological order, never future). This keeps the whole residual stage closed-form and lightweight.

### 4.4. Inference-Time Iterative Refinement

At test time, RGMR uses the full context window $W$ (70% of the series length). Let the resolution scale be $\mathcal{R} = \{r_1 > \cdots > r_K\}$ with $r_1$ the coarsest. We initialize

$$\begin{aligned} \bar{\mathbf{y}}^{(1)}_{t+1:t+H} &= f_\theta\big(\mathcal{P}_{r_1}(\mathbf{X}_{t-W+1:t})\big), \\ \widehat{\mathbf{y}}^{(1)}_{t+1:t+H} &= \bar{\mathbf{y}}^{(1)}_{t+1:t+H}. \end{aligned} \tag{3}$$

For each subsequent level $k = 2, \dots, K$, we query the frozen backbone at resolution $r_k$ to obtain a level-$k$ *raw (un-refined) forecast*, $\bar{\mathbf{y}}^{(k)}_{t+1:t+H} = f_\theta\big(\mathcal{P}_{r_k}(\mathbf{X}_{t-W+1:t})\big)$, where the bar marks a backbone proposal that has not yet been residual-corrected (in contrast to the refined $\widehat{\mathbf{y}}^{(k)}$ defined below; throughout, $\mathbf{y}$ is ground truth, $\widehat{\mathbf{y}}$ is the refined RGMR output, $\widetilde{\mathbf{y}}$ is the offline short-window cali-

bration proposal, and $\bar{\mathbf{y}}$ is the inference-time raw long-window proposal). We then form the *predicted* residual $\widehat{\mathbf{R}}_{t+1:t+H}^{(k-1)} = g_\phi^{(k-1)}(\mathbf{z}_t^{(k-1)})$ (no ground-truth residual is accessible at test time). We form adaptive elementwise weights

$$\boldsymbol{\omega}^{(k-1)} = \mathrm{clip}_{[\varepsilon_{\min},\,1]}\Big(\sigma\Big(\gamma\Big(\big|\widehat{\mathbf{R}}^{(k-1)}\big| - \delta^{(k-1)}\Big)\Big)\Big),$$
(4)

with $\sigma(x) = 1/(1+e^{-x})$. Each ingredient of Eq. (4) has a specific role: (i) the inner term $|\widehat{\mathbf{R}}^{(k-1)}| - \delta^{(k-1)}$ measures how far the predicted residual exceeds a validation-selected magnitude threshold $\delta^{(k-1)}$, so corrections only apply where prediction errors are likely large; (ii) $\sigma$ with small slope $\gamma$ (default $\gamma$=3.0) acts as a soft threshold that smoothly interpolates between "ignore" and "correct", avoiding hard cutoffs; (iii) $\mathrm{clip}_{[\varepsilon_{\min},1]}$ with $\varepsilon_{\min}$=$10^{-3}$ both prevents zero weights (which would freeze the forecast at the foundation prediction) and bounds weights by 1 (which prevents overcorrection). The threshold $\delta^{(k)}$ is a quantile of $|\widehat{\mathbf{R}}^{(k)}|$ selected on validation (grid $\{0.60,\ldots,0.90\}$; details in App. D).

The mixing coefficient is set by the conservative monotone schedule

$$\alpha^{(k)} = 0.3 + 0.5\Big(1 - \frac{r_k}{\max(\mathcal{R})}\Big) \in [0.3,\,0.8], \quad (5)$$

so that finer-resolution proposals receive a larger anchoring weight than coarser ones, while the correction term still receives at least 20% weight at the finest level. The constants 0.3 and 0.5 define a smooth coarse-to-fine ramp. We use a fixed step size $\eta^{(k)} \in (0,2]$ (default $\eta^{(k)}$=1.0 in the reported experiments); the analysis below states the corresponding non-expansive condition explicitly. Dropping the common horizon subscript $t+1{:}t+H$, we then update

$$\begin{aligned}
\widehat{\mathbf{y}}^{(k)} &= \alpha^{(k)}\,\bar{\mathbf{y}}^{(k)} \\
&+ \big(1-\alpha^{(k)}\big)\Big(\widehat{\mathbf{y}}^{(k-1)} + \eta^{(k)}\,\boldsymbol{\omega}^{(k-1)} \odot \widehat{\mathbf{R}}^{(k-1)}\Big).
\end{aligned}$$
(6)

This approach progressively corrects errors, integrating multi-resolution backbone proposals with level-specific residual corrections.

## 4.5. Theoretical Properties

We next analyze why RGMR can improve across refinement levels. Under bounded proposal error and bounded residual-prediction noise, each residual-guided update admits an assumption-conditional contraction bound: the previous expected squared error is multiplied by a factor below one, up to an additive noise floor. Our analysis differs from standard contraction and fixed-point treatments (Granas & Dugundji, 2003; Bai et al., 2019) by composing level-specific contraction operators and absorbing residual-prediction noise

as well as short-vs.-long window mismatch into a single bounded-second-moment error term. In particular, $\mathbf{e}^{(k-1)}$ below absorbs both the prediction noise of the Ridge regressor $g_\phi^{(k-1)}$ and any distribution shift between the short-window training residuals used to fit $g_\phi^{(k-1)}$ and the refined long-window residuals encountered at inference.

**Theorem 4.1** (Residual-guided contraction)**.** *Define the contraction factor*

$$\rho^{(k)} = (1-\alpha^{(k)}) \max_j \big|1 - \eta^{(k)}\boldsymbol{\omega}_j^{(k-1)}\big|. \quad (7)$$

*Suppose* $\widehat{\mathbf{R}}^{(k-1)} = \mathbf{R}^{(k-1)} + \mathbf{e}^{(k-1)}$, *where* $\mathbf{R}^{(k-1)} = \mathbf{y} - \widehat{\mathbf{y}}^{(k-1)}$ *is the analysis-only inference residual, with* $\mathbb{E}\|\mathbf{e}^{(k-1)}\|_2^2 < \infty$, *and suppose the level-$k$ proposal error* $\mathbf{u}^{(k)} = \mathbf{y} - \bar{\mathbf{y}}^{(k)}$ *has finite second moment. If* $\alpha^{(k)} \in (0,1)$, $\eta^{(k)} \in (0,2]$, *and* $\boldsymbol{\omega}_j^{(k-1)} \in [0,1]$, *then* $\rho^{(k)} < 1$ *and*

$$\begin{aligned}
\mathbb{E}\Big[\|\mathbf{y} - \widehat{\mathbf{y}}^{(k)}\|_2^2\Big] &\leq (1+\alpha^{(k)})(\rho^{(k)})^2\,\mathbb{E}\Big[\|\mathbf{y} - \widehat{\mathbf{y}}^{(k-1)}\|_2^2\Big] \\
&+ B_k.
\end{aligned}$$
(8)

*where* $B_k$ *collects bounded contributions from the level-$k$ proposal error and residual-prediction noise. Moreover, since* $\rho^{(k)} \leq 1 - \alpha^{(k)}$, *the leading factor satisfies* $(1+\alpha^{(k)})(\rho^{(k)})^2 \leq (1+\alpha^{(k)})(1-\alpha^{(k)})^2 < 1$, *ensuring a strict contraction at each level. (An explicit upper bound for* $B_k$ *is provided in the appendix.)*

**Corollary 4.2** (Finest-level no-harm envelope)**.** *Under the main setting* $r_K = 1$, *the direct finest-resolution baseline is exactly the base forecast of Section 3,* $\widehat{\mathbf{y}}^{\mathrm{base}} = \bar{\mathbf{y}}^{(K)} = f_\theta(\mathbf{X}_{t-W+1:t})$, *which performs no residual correction (no weights or step sizes). Assume* $\|\mathbf{y} - \bar{\mathbf{y}}^{(K)}\|_\infty \leq \bar{B}_K$ *and* $\|\mathbf{e}^{(K-1)}\|_\infty \leq E_{K-1}$. *The direct baseline then trivially satisfies* $\|\mathbf{y} - \widehat{\mathbf{y}}^{\mathrm{base}}\|_\infty \leq \bar{B}_K$. *If, in addition,*

$$\begin{aligned}
\max_{j\in[H]}\big|1 - \eta^{(K)}\,\boldsymbol{\omega}_j^{(K-1)}\big| \|\mathbf{y} - \widehat{\mathbf{y}}^{(K-1)}\|_\infty \\
+ \eta^{(K)}\|\boldsymbol{\omega}^{(K-1)}\|_\infty E_{K-1} \leq \bar{B}_K,
\end{aligned}$$
(9)

*then the refined prediction is inside the same $\ell_\infty$ envelope:*

$$\|\mathbf{y} - \widehat{\mathbf{y}}^{(K)}\|_\infty \leq \bar{B}_K.$$

## 4.6. Algorithm Description

To operationalize RGMR, we split the procedure into two phases: (1) *training* lightweight residual predictors using short-window contexts, and (2) *inference-time refinement* over multiple temporal resolutions with the frozen foundation model. Algorithm 1 summarizes the training stage, where residual predictors are learned from systematic errors identified on SPEI sequences.

Having obtained residual predictors, we apply them at inference time to refine multi-resolution forecasts generated by

**Algorithm 1:** *RGMR Training: Residual Predictor Learning*

---

**Input:** Training series $\{(\mathbf{X}_s, y_s)\}_{s=1}^{T_{\text{train}}}$, validation split, frozen model $f_\theta$, short window $L_{\text{short}}=12$, resolution scale $\mathcal{R} = \{r_1, \ldots, r_K\}$, horizon $H$

**Output:** Level-wise residual predictors and thresholds $\{g_\phi^{(k)}, \delta^{(k)}\}_{k=1}^{K-1}$ (applied at inference iteration $k+1$)

1   $\mathcal{D}^{(k)} \leftarrow \emptyset$ for all $k \in \{1, \ldots, K-1\}$;

2   **for** *each forecast origin $t$ in the training split with $t + H \leq T_{\text{train}}$* **do**

3      **for** $k = 1$ **to** $K-1$ **do**

4         $\widetilde{\mathbf{y}}_{t+1:t+H}^{(k)} \leftarrow f_\theta(\mathcal{P}_{r_k}(\mathbf{X}_{t-L_{\text{short}}+1:t}))$;

5         $\widehat{\mathbf{R}}_{t+1:t+H}^{(k)} \leftarrow \mathbf{y}_{t+1:t+H} - \widetilde{\mathbf{y}}_{t+1:t+H}^{(k)}$;

6         $\mathbf{z}_t^{(k)} \leftarrow$ [recent targets, rolling mean/std, OLS linear-trend slope, residual history];

7         $\mathcal{D}^{(k)} \leftarrow \mathcal{D}^{(k)} \cup \{(\mathbf{z}_t^{(k)}, \widetilde{\mathbf{R}}_{t+1:t+H}^{(k)})\}$;

8   **for** $k = 1$ **to** $K-1$ **do**

9      Fitting candidate Ridge predictors on the training split and evaluating them on the validation split;

10     Select $\delta^{(k)}$ on the validation split using validation predicted-residual magnitudes;

11     Fit the final $g_\phi^{(k)}$ on the union of the training and validation splits;

12   **return** $\{g_\phi^{(k)}, \delta^{(k)}\}_{k=1}^{K-1}$;

---

**Algorithm 2:** *RGMR Inference: Residual-Guided Multi-Resolution Refinement*

---

**Input:** Context $\mathbf{X}_{t-W+1:t}$; frozen TSFM $f_\theta$; residual predictors and thresholds $\{g_\phi^{(k)}, \delta^{(k)}\}_{k=1}^{K-1}$; resolution scale $\mathcal{R} = \{r_1, \ldots, r_K\}$; horizon $H$; mixing schedule $\alpha^{(k)}$ from Eq. (5)

**Output:** RGMR final prediction $\widehat{\mathbf{y}}_{t+1:t+H}^{(K)}$

1   $\bar{\mathbf{y}}^{(1)} \leftarrow f_\theta(\mathcal{P}_{r_1}(\mathbf{X}_{t-W+1:t}))$;   `// coarsest raw proposal`

2   $\widehat{\mathbf{y}}^{(1)} \leftarrow \bar{\mathbf{y}}^{(1)}$;     `// initialize refined forecast`

3   **for** $k \leftarrow 2$ **to** $K$ **do**

4      $\bar{\mathbf{y}}^{(k)} \leftarrow f_\theta(\mathcal{P}_{r_k}(\mathbf{X}_{t-W+1:t}))$; `// level-k raw base proposal`

5      $\mathbf{z}_t^{(k-1)} \leftarrow$ FeatExtract$(\mathbf{X}_{t-W+1:t}, \widehat{\mathbf{y}}^{(k-1)}, k-1, t)$;

6      $\widehat{\mathbf{R}}^{(k-1)} \leftarrow g_\phi^{(k-1)}(\mathbf{z}_t^{(k-1)})$;     `// predicted residual; no ground truth used`

7      $\boldsymbol{\omega}^{(k-1)} \leftarrow \text{clip}_{[\varepsilon_{\min}, 1]}\big(\sigma(\gamma(|\widehat{\mathbf{R}}^{(k-1)}| - \delta^{(k-1)}))\big)$; `// Eq. (4)`

8      Use fixed step size $\eta^{(k)}=1.0$ (any $\eta^{(k)} \in (0, 2]$ admissible by Thm. 4.1);

9      $\widehat{\mathbf{y}}^{(k)} \leftarrow \alpha^{(k)}\bar{\mathbf{y}}^{(k)} + (1-\alpha^{(k)})(\widehat{\mathbf{y}}^{(k-1)} + \eta^{(k)}\boldsymbol{\omega}^{(k-1)} \odot \widehat{\mathbf{R}}^{(k-1)})$;     `// Eq. (6)`

10   **return** $\widehat{\mathbf{y}}_{t+1:t+H}^{(K)}$;

---

the foundation model. Algorithm 2 describes this process: starting from the coarsest resolution, RGMR progressively integrates finer-resolution predictions, applies the predicted residual as a residual-guided correction, and uses the stability conditions stated in Theorem 4.1 to characterize the expected-error behavior under bounded-noise assumptions.

Together, Algorithms 1 and 2 instantiate the RGMR workflow: the training stage learns systematic level-specific biases on short windows, and the inference stage applies these biases as a residual-guided correction on top of long-window proposals. Under the stated stability and bounded-noise assumptions, this yields a contraction-style expected-error bound without retraining the base model.

### 4.7. Computational Complexity and Implementation

Let $K = |\mathcal{R}|$ denote the number of resolution levels (in our setup $K = 5$). Per refinement cycle, RGMR performs $K$ calls to the frozen foundation model $f_\theta$ (one per level), $K-1$ residual predictions using a lightweight Ridge regressor, and projection operations whose total cost is linear in the context length. The total cost per forecast origin is therefore

$$\mathcal{O}(K\, C_\theta + K\, W + (K-1)\, C_{\text{ridge}}), \qquad (10)$$

where $C_\theta$ is the cost of a single $f_\theta$ forward pass, $W$ is the input window length, and $C_{\text{ridge}} = \mathcal{O}(p_{\text{lag}} + q)$ is the per-step cost of residual feature evaluation (using $p_{\text{lag}}$ lags and a $q$-step trend) plus a closed-form Ridge prediction. In practice $C_{\text{ridge}} \ll C_\theta$, so the end-to-end runtime is dominated by the $K\, C_\theta$ term. The adaptive weighting and fixed step-size update contribute only element-wise operations that are negligible relative to $f_\theta$ calls.

RGMR operates entirely at inference time: there are *no parameter updates* to $f_\theta$, enabling immediate deployment on existing time-series foundation models without retraining. Hyperparameters are kept minimal and use conservative defaults (e.g., fixed resolution scale $\mathcal{R}$, weight-floor $\varepsilon_{\min}$, and step-size range $\eta \in (0, 2]$). When selection is required (e.g., residual-weight threshold quantiles $\delta^{(k)}$ or the Ridge penalty $\lambda^{(k)}$), we employ a *single* time-ordered validation pass; no test-time peeking or re-training of $f_\theta$ is involved. The adaptive weighting mechanism is implemented via element-wise sigmoids and simple clipping, and thus contributes negligible overhead compared to the $K$ forward calls of $f_\theta$.

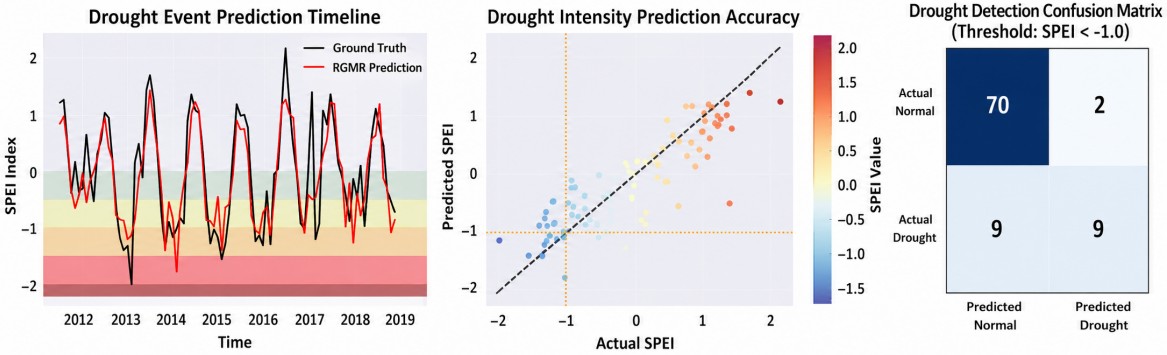

*Figure 3.* RGMR forecast quality and drought-event detection at a representative South Australia site. Left: monthly SPEI timeline with ground truth (black) and RGMR prediction (red); shaded bands indicate SPEI categories ranging from drought conditions at the bottom of the panel to wet conditions at the top. Middle: predicted vs. actual SPEI with $y=x$ (dashed); Pearson $r=0.83$; point color encodes SPEI. Right: confusion matrix for drought detection at threshold SPEI $< -1.0$ (precision 0.818, recall 0.500, F1 0.621); cell values are counts and colors reflect relative frequency.

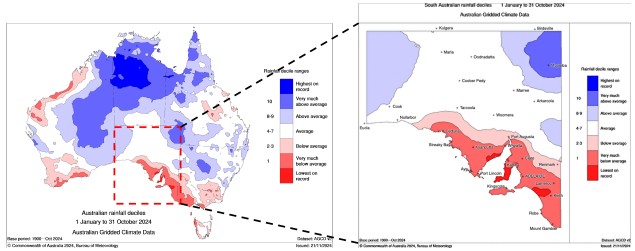

*Figure 4.* South Australia study-area context. The rainfall-decile map is used to illustrate regional rainfall heterogeneity and the 2024 drought context.

# 5. Experiments

We evaluate whether RGMR improves frozen foundation models for monthly-to-seasonal SPEI forecasting without parameter updates. The study targets overall accuracy against strong baselines, the effect of multi-resolution and residual refinement, the role of adaptive residual weighting, and computational overhead. We use rolling-origin evaluation with non-overlapping test windows (step size $s = H$); all standardization uses training-set statistics. Candidate residual predictors are fitted on the training split and selected on validation; after $\lambda^{(k)}$ and $\delta^{(k)}$ are fixed, the final residual predictors are refit on the union of training and validation splits and frozen before test evaluation. Test targets are never accessed at inference. Specifically, at each rolling origin $t$ the prediction uses only observations and inference residuals available up to month $t$; once $\mathbf{y}_{t+1:t+H}$ becomes observed, the corresponding residuals are appended to the residual-history feature store and may be consumed only by strictly later rolling origins, so no target inside the current horizon is used to form the current prediction. Datasets, regions, covariates, and splits are summarized in Appendix C; implementation details and hyperparameters

are in Appendix D. Code is available at `https://github.com/Wentao-Gao/RGMR_implementation`.

## 5.1. Setup

We evaluate regional SPEI forecasting across multiple nominal climate zones within South Australia (arid, temperate) and decades of records to ensure geographic and climatic diversity; exact regions, time spans, and sample counts are reported in Appendix C. Targets are SPEI at horizons $H$ months ahead; inputs are multivariate climate covariates and past SPEI within a window of length $W$.

**Study area.** We focus on South Australia (as shown in Figure 4) as the study area for two main reasons. First, the region is highly vulnerable to drought and interannual rainfall variability, making seasonal prediction societally important. Second, reliable meteorological records (NCEP–NCAR Reanalysis 1 data (Kalnay et al., 1996)) are available for this region, ensuring reproducibility. Recent reports show that parts of South Australia recorded their driest year on record in 2024, with rainfall totals at multiple locations reaching historic lows between June 2024 and May 2025 (ABC News, 2025b;a). Such severe and prolonged droughts underline the acute need for accurate climate forecasting in the region.

**Baselines and protocol.** We compare against frozen foundation models, multi-resolution architectures, and generic deep baselines. *Foundation-only* includes TimesFM (Das et al., 2024), TimeGPT (Garza et al., 2023), and TabPFN (Prior Labs, 2025) on the raw window. *Multi-resolution* baselines include N-BEATS (Oreshkin et al., 2020) (5 stacks with standard trend/seasonality blocks) and Scaleformer (Shabani et al., 2023) with learned scale selection. *Generic deep* baselines used in Table 1 include Transformer (Vaswani et al., 2017), Autoformer (Wu et al., 2021), Crossformer (Zhang &

*Table 1.* One-month-ahead SPEI forecasting at three South Australian locations. Best results in **bold**.

| Method | Location 1 $(-26.125°, 129.125°)$ | | | Location 2 $(-29.125°, 134.875°)$ | | | Location 3 $(-35.625°, 138.875°)$ | | |
| --- | --- | --- | --- | --- | --- | --- | --- | --- | --- |
| | MSE↓ | MAE↓ | R²↑ | MSE↓ | MAE↓ | R²↑ | MSE↓ | MAE↓ | R²↑ |
| *Multi-resolution baselines* | | | | | | | | | |
| N–BEATS (5 stacks) | 0.482 | 0.497 | 0.505 | 0.439 | 0.525 | 0.589 | 0.668 | 0.590 | 0.409 |
| Scaleformer | 0.472 | 0.491 | 0.510 | 0.445 | 0.520 | 0.582 | 0.662 | 0.585 | 0.413 |
| *Generic deep baselines* | | | | | | | | | |
| Transformer | 0.537 | 0.554 | 0.451 | 0.510 | 0.581 | 0.522 | 0.745 | 0.655 | 0.365 |
| Autoformer | 0.546 | 0.559 | 0.442 | 0.547 | 0.582 | 0.487 | 0.752 | 0.658 | 0.358 |
| Crossformer | 0.542 | 0.557 | 0.446 | 0.474 | 0.588 | 0.556 | 0.740 | 0.654 | 0.370 |
| DLinear | 0.533 | 0.551 | 0.455 | 0.423 | 0.553 | 0.604 | 0.735 | 0.650 | 0.375 |
| FiLM | 0.558 | 0.569 | 0.430 | 0.529 | 0.597 | 0.505 | 0.760 | 0.662 | 0.350 |
| iTransformer | 0.524 | 0.546 | 0.464 | 0.454 | 0.539 | 0.575 | 0.738 | 0.652 | 0.372 |
| PatchTST | 0.535 | 0.552 | 0.453 | 0.461 | 0.548 | 0.568 | 0.743 | 0.655 | 0.366 |
| TimesNet | 0.550 | 0.562 | 0.438 | 0.489 | 0.572 | 0.542 | 0.754 | 0.659 | 0.356 |
| TSMixer | 0.529 | 0.549 | 0.459 | 0.468 | 0.555 | 0.561 | 0.736 | 0.651 | 0.374 |
| *Foundation models and RGMR* | | | | | | | | | |
| TabPFN-ts v1.0 | 0.462 | 0.503 | 0.521 | 0.389 | 0.445 | 0.632 | 0.566 | 0.562 | 0.442 |
| TabPFN-ts v1.0 + RGMR | 0.344 | 0.451 | 0.615 | 0.305 | 0.401 | 0.713 | 0.452 | 0.489 | 0.541 |
| TimeGPT-1 | 0.437 | 0.489 | 0.536 | 0.351 | 0.431 | 0.658 | 0.543 | 0.551 | 0.456 |
| TimeGPT-1 + RGMR | 0.325 | 0.437 | 0.632 | 0.278 | 0.386 | 0.738 | 0.431 | 0.475 | 0.562 |
| TimesFM v1.0 | 0.391 | 0.465 | 0.564 | 0.318 | 0.412 | 0.687 | 0.524 | 0.536 | 0.469 |
| TimesFM v1.0 + RGMR | **0.318** | **0.423** | **0.651** | **0.258** | **0.334** | **0.746** | **0.426** | **0.436** | **0.568** |

*Table 2.* Ablation of RGMR components on one-month-ahead SPEI forecasting (averaged across three South Australian sites).

| Method | TabPFN-ts v1.0 | | | TimeGPT-1 | | | TimesFM v1.0 | | |
| --- | --- | --- | --- | --- | --- | --- | --- | --- | --- |
| | MSE↓ | MAE↓ | R²↑ | MSE↓ | MAE↓ | R²↑ | MSE↓ | MAE↓ | R²↑ |
| Foundation Model (frozen) | 0.472 | 0.503 | 0.532 | 0.444 | 0.490 | 0.550 | 0.411 | 0.471 | 0.573 |
| + Coarse projection only | 0.436 | 0.482 | 0.570 | 0.410 | 0.467 | 0.590 | 0.387 | 0.448 | 0.604 |
| + Multi-resolution (no refinement) | 0.414 | 0.469 | 0.592 | 0.390 | 0.455 | 0.615 | 0.372 | 0.433 | 0.627 |
| + RGMR (no weights) | 0.380 | 0.459 | 0.609 | 0.360 | 0.442 | 0.625 | 0.341 | 0.405 | 0.646 |
| + RGMR (Full) | **0.367** | **0.447** | **0.623** | **0.345** | **0.433** | **0.644** | **0.334** | **0.398** | **0.655** |

Yan, 2023), DLinear (Zeng et al., 2023), FiLM (Zhou et al., 2022), iTransformer (Liu et al., 2024b), PatchTST (Nie et al., 2023), TimesNet (Wu et al., 2023), and TSMixer (Chen et al., 2023). Unless stated otherwise we use the public reference implementations released with each method, gathered through the THUML Time-Series-Library (Tsinghua Machine Learning Group, 2024). All methods use identical inputs, splits, and preprocessing; none updates $f_\theta$ during inference. We report MSE, MAE, and $R^2$ per site.

### 5.2. Main Results

#### 5.2.1. ONE-MONTH-AHEAD SPEI

Figure 3 provides an overview of RGMR performance on South Australian SPEI, illustrating both forecasting quality and downstream drought-event detection. Table 1 reports one-month-ahead results at three South Australian locations, including N-BEATS and Scaleformer as multi-resolution baselines.

Foundation models (TabPFN, TimeGPT, TimesFM) already outperform generic deep baselines, and applying RGMR on top of TimesFM yields the strongest performance across the reported metrics and sites. Specifically, MSE decreases from 0.391 to **0.318** at Location 1, from 0.318 to **0.258** at Location 2, and from 0.524 to **0.426** at Location 3. $R^2$ improves consistently.

Compared directly with multi-resolution baselines, TimesFM+RGMR achieves lower MSE than both Scaleformer and N-BEATS at all sites, with similar gaps across other metrics. These results highlight RGMR's ability to deliver consistent gains beyond both generic deep baselines and established multi-resolution architectures.

Table 2 averages results over the three locations and decomposes RGMR into its components. Using a single coarse resolution improves the foundation baselines, multi-resolution without residual refinement brings further gains, residual-guided updates add monotonic improvements, and adaptive residual weights attain the best overall performance. This progression is consistent with our design choices: under the

bounded-noise assumptions in Theorem 4.1, the per-level update achieves contraction with modulus $\rho^{(k)} < 1$ when $\eta^{(k)} \in (0, 2]$ and $\omega_j^{(k-1)} \in [0, 1]$, and the multi-level behavior exhibits geometric decay to a noise floor in line with the contraction analysis.

### 5.3. Generalization

We further evaluate RGMR from three complementary perspectives: refinement behavior, computational cost, and sensitivity to resolution design, with additional experiments provided in Appendix G, in order to assess both effectiveness and practicality.

#### 5.3.1. SAMPLE-WISE IMPROVEMENT DIAGNOSTIC

*Table 3.* Per-level sample-wise improvement diagnostic on the South Australian held-out test set. The "Sample-wise improvement ratio" counts how often the per-sample squared error strictly decreases from refinement level $k-1$ to $k$.

| Refinement level $k$ | Resolution stride $r_k$ | Sample-wise improvement ratio |
|---|---|---|
| 2 | 6 | 39.7% |
| 3 | 3 | 79.4% |
| 4 | 2 | 66.2% |
| 5 | 1 | 48.5% |

To empirically examine the refinement behavior we compute, for each refinement level $k$, the fraction of test origins whose squared error *strictly* decreases from level $k-1$ to level $k$,

$$\frac{1}{N} \sum_{i=1}^{N} \mathbf{1}\big\{ \big\| \mathbf{y}_i - \widehat{\mathbf{y}}_i^{(k)} \big\|_2^2 < \big\| \mathbf{y}_i - \widehat{\mathbf{y}}_i^{(k-1)} \big\|_2^2 \big\}.$$

The ratios in Table 3 fall between roughly 40% and 80%, with the largest jump at $k=3$. Two clarifications matter. First, at the finest stride $r_5=1$ the level-4 prediction is already close to ground truth on many easy samples, so further refinement is mostly noise-bound; the 48.53% sample-wise rate is therefore consistent with, not a violation of, the theorem. Second, Theorem 4.1 controls the *expected* squared error in $\ell_2$, which is what our aggregated MSE in Tables 1 and 2 verifies, while Table 3 is a per-sample diagnostic.

#### 5.3.2. LATENCY AND COMPUTATIONAL EFFICIENCY

Table 4 reports GPU latency measured within the command `torch.cuda.Event`. Despite multiple forward passes, RGMR introduces only ∼60 ms additional overhead, which is negligible for monthly drought forecasting.

#### 5.3.3. SENSITIVITY TO RESOLUTION SCALE CHOICE

Table 5 compares five resolution scale variants. RGMR achieves stable performance across all five variants, indicating low sensitivity to the specific resolution-scale choice.

*Table 4.* GPU latency per rolling forecast origin on a single RTX 3090.

| Model | Mean (ms) | Std (ms) |
|---|---|---|
| TimesFM v1.0 | 14.52 | 2.18 |
| TimesFM v1.0 + RGMR | 74.89 | 5.00 |

*Table 5.* Resolution-scale sensitivity at Location 1. We use "resolution scale" for the ordered set of strides $\mathcal{R}$ used during refinement, "resolution level" for an individual element of $\mathcal{R}$ (indexed $k = 1, \ldots, K$ from coarsest to finest), and "stride" for the numerical value $r_k$, consistent with Table 6 (Appendix A.1) and Table 3.

| Variant | Resolution scale | MSE↓ | MAE↓ | $R^2$↑ |
|---|---|---|---|---|
| S1 | [12,6,3,2,1] | **0.318** | **0.423** | **0.651** |
| S2 | [16,8,4,2,1] | 0.324 | 0.439 | 0.644 |
| S3 | [12,4,2,1] | 0.322 | 0.438 | 0.645 |
| S4 | [12,6,3,1] | 0.330 | 0.443 | 0.637 |
| S5 | [12,6,4,3,1] | 0.330 | 0.443 | 0.636 |

## 6. Conclusion

This paper introduced RGMR, a residual-guided multi-resolution refinement framework that adapts frozen time series foundation models to regional drought forecasting. RGMR iteratively projects predictions across multiple temporal scales and corrects them with predicted, test-time, ground-truth-free residuals, allowing the model to capture long-term climate structure and short-term variability without updating base model parameters. Under the stated stability and bounded-noise assumptions, our analysis gives a contraction-style bound on the expected squared point-forecast error across refinement levels until reaching a noise-determined floor. The strong benchmark performance suggests that RGMR may offer a general and practical direction for adapting frozen time-series foundation models to broader regional climate forecasting tasks.

## Impact Statement

This paper aims to advance the use of frozen time-series foundation models for regional climate forecasting, specifically drought monitoring via SPEI. The most direct positive consequences are in (i) water-allocation and reservoir-operation planning under prolonged dry spells, (ii) seasonal agricultural and irrigation scheduling in water-limited regions, (iii) drought early-warning systems for emergency-resource preparation, and (iv) supporting adaptation planning for regional and remote communities most exposed to climate variability. Because the wrapper does not modify the foundation backbone, it can be retrofitted into existing operational pipelines at low cost. We also acknowledge that any operational forecasting tool can be misused if treated as the sole basis for high-stakes water-rights, insurance, or emergency-response decisions; we therefore advocate using RGMR as one input to a multi-evidence decision process, not as a standalone authority.

## Acknowledgments

This work was supported by the ARC Discovery Project DP230101122 and the Adelaide University Research Training Program (RTP) Scholarship. We gratefully acknowledge the continued support from the CSIRO Environment Research Unit and the Data61 Business Unit.

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

# A. Notation and Projection Operators

## A.1. Notation

Table 6 summarizes the main notation used in the paper; auxiliary symbols introduced only locally (e.g., $T_{\mathrm{train}}$, $\mathcal{D}^{(k)}$, $C_\theta$, and $C_{\mathrm{ridge}}$) are defined at their first use in the algorithms or complexity discussion. Conventions: dimensions are stated explicitly; level-indexed prediction and residual quantities are vectors in $\mathbb{R}^H$ at a fixed forecast origin $t$.

*Table 6.* Summary of the main notation used in the paper.

| Symbol | Meaning |
|---|---|
| *Inputs, targets, and backbone* | |
| $T$ | length of the full SPEI series (months) |
| $W$ | full inference context length passed to $f_\theta$ |
| $L_{\mathrm{short}}$ | short context length used during offline calibration to fit residual predictors (default 12 months) |
| $D$ | feature dimension of $\mathbf{X}_t$ (past SPEI plus climate covariates) |
| $H$ | forecast horizon ($H{=}1$ in the main results) |
| $\mathbf{X}_t \in \mathbb{R}^D$ | input vector at time $t$; first channel is the historical SPEI value $y_t$, remaining $D{-}1$ channels are climate covariates |
| $\mathbf{X}_{t-W+1:t} \in \mathbb{R}^{W \times D}$ | multivariate input window ending at the forecast origin $t$ (training-split standardization) |
| $\mathbf{y}_{t+1:t+H} \in \mathbb{R}^H$ | target SPEI vector for the next $H$ months |
| $f_\theta$ | frozen TSFM backbone (no parameter updates at inference) |
| *Resolution scale and projections* | |
| $\mathcal{R} = \{r_1 > \cdots > r_K\}$ | resolution scale from coarse to fine; $K = |\mathcal{R}|$, $r_K{=}1$ is the native finest level (main setting: $\mathcal{R} = \{12, 6, 3, 2, 1\}$) |
| $\mathcal{D}_r, \mathcal{U}_r$ | block-average down-projection / repeat up-projection at stride $r$ |
| $\mathcal{P}_r = \mathcal{U}_r \circ \mathcal{D}_r$ | projection operator at stride $r$ (used for both training and inference proposals) |
| *Predictions and residuals (training- vs. inference-time)* | |
| $\bar{\mathbf{y}}^{(k)} \in \mathbb{R}^H$ | level-$k$ *raw (un-refined) base proposal* at inference: $\bar{\mathbf{y}}^{(k)} = f_\theta(\mathcal{P}_{r_k}(\mathbf{X}_{t-W+1:t}))$; the bar marks a backbone proposal that has not been residual-corrected |
| $\widehat{\mathbf{y}}^{(k)} \in \mathbb{R}^H$ | *refined* prediction at inference after refinement level $k$, with $\widehat{\mathbf{y}}^{(1)}{=}\bar{\mathbf{y}}^{(1)}$; the final RGMR output is $\widehat{\mathbf{y}}^{(K)}$ |
| $\widehat{\mathbf{y}}^{\mathrm{base}} \in \mathbb{R}^H$ | single-pass base forecast from $f_\theta$ on the full $W$-context: $\widehat{\mathbf{y}}^{\mathrm{base}} = f_\theta(\mathbf{X}_{t-W+1:t})$; introduced in Section 3, and (since $r_K{=}1$ in the main setting) it coincides with $\bar{\mathbf{y}}^{(K)}$ and serves as the no-RGMR direct baseline in Corollary 4.2 |
| $\widetilde{\mathbf{y}}^{(k)} \in \mathbb{R}^H$ | short-window proposal computed during offline calibration: $\widetilde{\mathbf{y}}^{(k)} = f_\theta(\mathcal{P}_{r_k}(\mathbf{X}_{t-L_{\mathrm{short}}+1:t}))$ |
| $\widetilde{\mathbf{R}}^{(k)} \in \mathbb{R}^H$ | *training* residual: $\widetilde{\mathbf{R}}^{(k)} = \mathbf{y} - \widetilde{\mathbf{y}}^{(k)}$ (used before test evaluation; uses observed ground truth only from the training/validation periods) |
| $\widehat{\mathbf{R}}^{(k)} \in \mathbb{R}^H$ | *predicted* residual at inference: $\widehat{\mathbf{R}}^{(k)} = g_\phi^{(k)}(\mathbf{z}_t^{(k)})$. The only residual quantity available at test time; ground truth $\mathbf{y}_{t+1:t+H}$ is never accessed. |
| $\mathbf{R}^{(k)} \in \mathbb{R}^H$ | inference residual w.r.t. ground truth: $\mathbf{R}^{(k)} = \mathbf{y} - \widehat{\mathbf{y}}^{(k)}$; an analysis-only quantity and the target of the residual predictor $g_\phi^{(k)}$. Never accessed by the algorithm at test time. |
| $\mathbf{e}^{(k)} \in \mathbb{R}^H$ | residual-prediction error of $g_\phi^{(k)}$: $\widehat{\mathbf{R}}^{(k)} = \mathbf{R}^{(k)} + \mathbf{e}^{(k)}$, zero-mean with bounded second moment; it includes Ridge prediction noise and short-vs.-long window mismatch |
| $\mathbf{u}^{(k)} \in \mathbb{R}^H$ | level-$k$ proposal error: $\mathbf{u}^{(k)} = \mathbf{y} - \bar{\mathbf{y}}^{(k)}$ |
| $g_\phi^{(k)}$, $\boldsymbol{\Theta}^{(k)}$ | level-$k$ Ridge residual predictor and its coefficient matrix $\boldsymbol{\Theta}^{(k)} \in \mathbb{R}^{d_z \times H}$; fitted on the training split for validation selection and refit on training+validation before test inference |
| $\mathbf{z}_t^{(k)} \in \mathbb{R}^{d_z}$ | feature vector for the level-$k$ residual predictor (written $\mathbf{z}_t$ when $k$ is clear): concatenation of (i) most recent targets, (ii) rolling mean/std, (iii) OLS linear-trend slope, and (iv) most recent inference-residual lags |
| $p_{\mathrm{lag}}$ | number of lagged target and residual values used in $\mathbf{z}_t^{(k)}$ (default 6) |
| *Update parameters and stability quantities* | |
| $\boldsymbol{\omega}^{(k)} \in [0, 1]^H$ | adaptive elementwise residual weight vector at level $k$: $\boldsymbol{\omega}^{(k)} = \mathrm{clip}_{[\varepsilon_{\min}, 1]}(\sigma(\gamma(|\widehat{\mathbf{R}}^{(k)}| - \delta^{(k)})))$ |
| $\gamma$ | sigmoid sharpness (small constant, default 3.0) |
| $\delta^{(k)}$ | validation-selected quantile of $|\widehat{\mathbf{R}}^{(k)}|$ used as the soft-threshold midpoint |
| $\varepsilon_{\min}$ | weight floor for numerical stability (default $10^{-3}$) |
| $\alpha^{(k)} \in (0, 1)$ | mixing coefficient at level $k$ (monotone increasing in $1/r_k$; main schedule $\alpha^{(k)} = 0.3 + 0.5(1 - r_k/\max \mathcal{R})$) |
| $\eta^{(k)} \in (0, 2]$ | fixed step size at level $k$ (default 1.0 in the reported experiments) |
| $\rho^{(k)}$ | contraction factor at level $k$, $\rho^{(k)} = (1 - \alpha^{(k)}) \max_j |1 - \eta^{(k)} \boldsymbol{\omega}_j^{(k-1)}|$; *induced* by $\alpha, \eta, \boldsymbol{\omega}$ rather than separately tuned |
| $B_k$ | bounded additive error term at level $k$ (proposal error and residual-prediction noise; explicit upper bound in App. B) |
| $\bar{B}_K$ | deterministic $\ell_\infty$ upper bound on the level-$K$ backbone error, $\|\mathbf{y} - \bar{\mathbf{y}}^{(K)}\|_\infty \leq \bar{B}_K$; used in the finest-level no-harm envelope of Corollary 4.2 (distinct from the expectation-$\ell_2$ quantity $B_k$ in Theorem 4.1) |
| $\sigma_e^2$ | validation upper bound on $\mathbb{E}\|\mathbf{e}^{(k)}\|_2^2$ (residual-prediction noise floor) |
| $E_{K-1}$ | uniform $\ell_\infty$ bound on $\mathbf{e}^{(K-1)}$ used in Corollary 4.2 |
| $\boldsymbol{\Omega}^{(k)}$ | $\mathrm{Diag}(\boldsymbol{\omega}^{(k)})$ |

**Conventions.** $\|\cdot\|_2$ is the Euclidean norm for vectors and the spectral (operator) norm for matrices. $\odot$ is the elementwise (Hadamard) product. $\mathrm{Diag}(\mathbf{v})$ forms a diagonal matrix from the entries of $\mathbf{v}$. $\sigma(x) = 1/(1 + e^{-x})$ is the standard logistic.

*Unification note.* The validation-selected residual-magnitude threshold is denoted $\delta^{(k)}$ throughout the paper; the logistic uses $\sigma(x) = 1/(1 + e^{-x})$ with fixed sharpness $\gamma = 3.0$; weights are clipped elementwise to $[\varepsilon_{\min}, 1]$ with $\varepsilon_{\min} > 0$ by default.

### A.2. Projection used in the main experiments (simple average & repeat)

All main results use the simple projection

$$\mathcal{P}_r = \mathcal{U}_r \circ \mathcal{D}_r,$$

where $\mathcal{D}_r$ averages non-overlapping blocks of length $r$ and $\mathcal{U}_r$ repeats each block average $r$ times. If the context length is not divisible by $r$, the final partial block is averaged over its available entries and repeated only up to the original sequence length, so no future padding is used. All channels share the same timing to preserve cross-channel coherence. This projection is deterministic, lightweight ($O(W)$ per level), and is used *throughout the paper and all ablations* unless stated otherwise.

**Lemma A.1** (Boundedness of the simple projection). *Let $\mathcal{D}_r$ be block-averaging (row-stochastic) and $\mathcal{U}_r$ be repeat-upsampling. Then $\|\mathcal{D}_r\|_2 \leq 1$, $\|\mathcal{U}_r\|_2 \leq \sqrt{r}$, and hence $\|\mathcal{P}_r\|_2 = \|\mathcal{U}_r\mathcal{D}_r\|_2 \leq \|\mathcal{U}_r\|_2\|\mathcal{D}_r\|_2 \leq \sqrt{r}$. Moreover, because $\mathcal{P}_r$ replaces each block by its within-block mean, it is an idempotent orthogonal projection onto the block-constant subspace, and therefore $\|\mathcal{P}_r\|_2 \leq 1$.*

## B. Proofs and Technical Details

### B.1. RGMR update and residual recursion

Throughout this appendix, $K$ denotes the final refinement level (corresponding to the finest resolution stride $r_K = 1$). We restate the RGMR update from Section 4 (suppressing the origin index $t$ for clarity):

$$\widehat{\mathbf{y}}^{(k)} = \alpha^{(k)}\bar{\mathbf{y}}^{(k)} + (1 - \alpha^{(k)})\Big(\widehat{\mathbf{y}}^{(k-1)} + \eta^{(k)}\boldsymbol{\omega}^{(k-1)} \odot \widehat{\mathbf{R}}^{(k-1)}\Big), \qquad k \geq 2, \quad \widehat{\mathbf{y}}^{(1)} = \bar{\mathbf{y}}^{(1)}. \tag{11}$$

For the analysis, we view the frozen residual predictor used at level $k-1$ as an estimator of the current inference residual; any mismatch between the short-window residuals used to train the Ridge predictor and the long-window refined residual at inference is included in the residual-prediction error term. Here the predicted residual is decomposed as $\widehat{\mathbf{R}}^{(k-1)} = \mathbf{R}^{(k-1)} + \mathbf{e}^{(k-1)}$ with $\mathbf{R}^{(k-1)} = \mathbf{y} - \widehat{\mathbf{y}}^{(k-1)}$ and $\mathbf{e}^{(k-1)}$ denoting residual-prediction error (zero mean, bounded second moment). Let $\mathbf{\Omega}^{(k-1)} = \mathrm{Diag}(\boldsymbol{\omega}^{(k-1)})$. Subtracting (11) from $\mathbf{y}$ yields the residual recursion

$$\mathbf{R}^{(k)} = (1 - \alpha^{(k)})\big(I - \eta^{(k)}\mathbf{\Omega}^{(k-1)}\big)\mathbf{R}^{(k-1)} - (1 - \alpha^{(k)})\eta^{(k)}\mathbf{\Omega}^{(k-1)}\mathbf{e}^{(k-1)} + \alpha^{(k)}\underbrace{(\mathbf{y} - \bar{\mathbf{y}}^{(k)})}_{\mathbf{u}^{(k)}}. \tag{12}$$

**Lemma B.1** (Diagonal operator norm identity). *If $D$ is diagonal with entries in $[0, 1]$ and $\eta > 0$, then $\big\|I - \eta D\big\|_2 = \max_j |1 - \eta D_{jj}|$.*

*Proof.* $I - \eta D$ is diagonal; its spectral norm equals the largest absolute diagonal entry. $\qquad\square$

### B.2. Proof of Theorem 4.1

Define
$$M^{(k)} = (1 - \alpha^{(k)})(I - \eta^{(k)}\mathbf{\Omega}^{(k-1)}), \quad N^{(k)} = -(1 - \alpha^{(k)})\eta^{(k)}\mathbf{\Omega}^{(k-1)}, \quad v^{(k)} = \alpha^{(k)}\mathbf{u}^{(k)}.$$

Then (12) can be written as $\mathbf{R}^{(k)} = M^{(k)}\mathbf{R}^{(k-1)} + N^{(k)}\mathbf{e}^{(k-1)} + v^{(k)}$. In the following, expectations are taken over the residual-prediction error $\mathbf{e}^{(k-1)}$, conditional on the fixed context and the resulting adaptive weights at the forecast origin; hence $M^{(k)}, N^{(k)}, v^{(k)}$ are treated as fixed inside $\mathbb{E}[\cdot]$.

By Lemma B.1,

$$\|M^{(k)}\|_2 = (1 - \alpha^{(k)})\big\|I - \eta^{(k)}\mathbf{\Omega}^{(k-1)}\big\|_2 = (1 - \alpha^{(k)})\max_j |1 - \eta^{(k)}\boldsymbol{\omega}_j^{(k-1)}| =: \rho^{(k)}.$$

With $\alpha^{(k)} \in (0,1)$, $\eta^{(k)} \in (0,2]$, and $\omega_j^{(k-1)} \in [0,1]$, we have $|1 - \eta^{(k)} \omega_j^{(k-1)}| \le 1$, hence $\rho^{(k)} \le 1 - \alpha^{(k)} < 1$.

Next, use the two-parameter Young inequality twice: for any $\delta_1, \delta_2 > 0$ and vectors $u, v, w$,

$$\|u + v + w\|_2^2 \le (1 + \delta_1)\|u\|_2^2 + (1 + \tfrac{1}{\delta_1})(1 + \delta_2)\|v\|_2^2 + (1 + \tfrac{1}{\delta_1})(1 + \tfrac{1}{\delta_2})\|w\|_2^2.$$

Apply this to $u = M^{(k)}\mathbf{R}^{(k-1)}$, $v = N^{(k)}\mathbf{e}^{(k-1)}$, $w = v^{(k)}$, take expectations, and bound norms:

$$\mathbb{E}\|\mathbf{R}^{(k)}\|_2^2 \le (1 + \delta_1)\|M^{(k)}\|_2^2 \, \mathbb{E}\|\mathbf{R}^{(k-1)}\|_2^2 + (1 + \tfrac{1}{\delta_1})(1 + \delta_2)\|N^{(k)}\|_2^2 \, \mathbb{E}\|\mathbf{e}^{(k-1)}\|_2^2$$
$$+ (1 + \tfrac{1}{\delta_1})(1 + \tfrac{1}{\delta_2})\|v^{(k)}\|_2^2.$$

**Explicit, level-adaptive constants.** Choose $\boxed{\delta_1 = \alpha^{(k)}, \ \delta_2 = 1}$. Then

$$(1 + \delta_1)\|M^{(k)}\|_2^2 = (1 + \alpha^{(k)})(\rho^{(k)})^2 \ \le \ (1 + \alpha^{(k)})(1 - \alpha^{(k)})^2 = 1 - \alpha^{(k)} - (\alpha^{(k)})^2 + (\alpha^{(k)})^3 < 1,$$

so the leading factor remains a strict contraction. Moreover,

$$\|N^{(k)}\|_2 \le (1 - \alpha^{(k)})\eta^{(k)}\|\mathbf{\Omega}^{(k-1)}\|_2 \le (1 - \alpha^{(k)})\eta^{(k)}, \qquad \|v^{(k)}\|_2 = \alpha^{(k)}\|\mathbf{u}^{(k)}\|_2.$$

Hence we obtain

$$\mathbb{E}\|\mathbf{R}^{(k)}\|_2^2 \ \le \ \underbrace{\left[(1 + \alpha^{(k)})(\rho^{(k)})^2\right]}_{< 1} \mathbb{E}\|\mathbf{R}^{(k-1)}\|_2^2 \ + \ B_k, \tag{13}$$

with the explicit bound

$$B_k \ \le \ 2\left(1 + \tfrac{1}{\alpha^{(k)}}\right)(1 - \alpha^{(k)})^2 (\eta^{(k)})^2 \sigma_e^2 \ + \ 2\left(1 + \tfrac{1}{\alpha^{(k)}}\right)(\alpha^{(k)})^2 \, \mathbb{E}\|\mathbf{u}^{(k)}\|_2^2, \tag{14}$$

where $\sigma_e^2$ upper-bounds the *conditional* second moment of $\mathbf{e}^{(k-1)}$ (estimated on validation), i.e., $\sigma_e^2 \ge \mathbb{E}\|\mathbf{e}^{(k-1)}\|_2^2$ under the above conditioning. This proves Theorem 4.1 with fully spelled-out constants.

**Modeling note (Ridge $\Rightarrow$ bounded noise).** In code, $\mathbf{e}^{(k-1)}$ is precisely the prediction error of the (deterministic) Ridge regressor $g_\phi^{(k-1)}$. Treating it as zero-mean with bounded second moment is a standard simplifying assumption for contraction-style analyses; the variance proxy $\sigma_e^2$ is estimated on validation.

### B.3. Proof of Corollary 4.2

*Proof.* We proceed in three steps.

**Step 1: Invoke the per-level bound at level $K$.** By the update rule and the non-expansive step-size condition in Section 4.5, the level-$K$ prediction satisfies the per-level $\ell_\infty$ bound

$$\|\mathbf{y} - \widehat{\mathbf{y}}^{(K)}\|_\infty \le (1 - \alpha^{(K)}) \underbrace{\max_{j \in [H]} \left|1 - \eta^{(K)} \omega_j^{(K-1)}\right|}_{\text{non-expansive factor}} \|\mathbf{y} - \widehat{\mathbf{y}}^{(K-1)}\|_\infty$$
$$+ \ \alpha^{(K)} \bar{B}_K \ + \ (1 - \alpha^{(K)})\eta^{(K)} \|\boldsymbol{\omega}^{(K-1)}\|_\infty E_{K-1}. \tag{15}$$

This is exactly the $k{=}K$ instance of the general per-level inequality in Section 4.5, obtained by: (i) subtracting $\mathbf{y}$ from the level-$K$ update, (ii) decomposing the residual prediction as $\widehat{\mathbf{R}}^{(K-1)} = \mathbf{R}^{(K-1)} + \mathbf{e}^{(K-1)}$ with $\|\mathbf{e}^{(K-1)}\|_\infty \le E_{K-1}$, (iii) applying the operator norm identity $\|I - \eta^{(K)}\mathrm{Diag}(\boldsymbol{\omega}^{(K-1)})\|_{\infty \to \infty} = \max_j |1 - \eta^{(K)}\omega_j^{(K-1)}|$, and (iv) using $\|\mathbf{a} \odot \mathbf{b}\|_\infty \le \|\mathbf{a}\|_\infty \|\mathbf{b}\|_\infty$ together with the backbone error bound $\|\mathbf{y} - \bar{\mathbf{y}}^{(K)}\|_\infty \le \bar{B}_K$.

**Step 2: Upper bound for the direct finest-level baseline.** By the definition of $\bar{B}_K$ (the $\ell_\infty$ upper bound on the level-$K$ backbone error), the direct one-pass baseline obeys

$$\|\mathbf{y} - \widehat{\mathbf{y}}^{\text{base}}\|_\infty = \|\mathbf{y} - \bar{\mathbf{y}}^{(K)}\|_\infty \ \le \ \bar{B}_K. \tag{16}$$

**Step 3: Sufficient condition for the refined envelope.** Starting from (15), it is sufficient that the remaining two terms are together no larger than $\bar{B}_K$, i.e.,

$$\max_{j \in [H]} \left| 1 - \eta^{(K)} \omega_j^{(K-1)} \right| \|\mathbf{y} - \widehat{\mathbf{y}}^{(K-1)}\|_\infty + \eta^{(K)} \|\boldsymbol{\omega}^{(K-1)}\|_\infty E_{K-1} \leq \bar{B}_K. \tag{17}$$

Under (17), inequality (15) yields

$$\|\mathbf{y} - \widehat{\mathbf{y}}^{(K)}\|_\infty \leq \alpha^{(K)} \bar{B}_K + \left( \bar{B}_K - \alpha^{(K)} \bar{B}_K \right) = \bar{B}_K,$$

which, together with (16), gives the envelope claim of Corollary 4.2. □

### B.4. Complexity and bound on $\mathcal{P}_r$

For $\mathcal{P}_r = \mathcal{U}_r \circ \mathcal{D}_r$ (block average + repeat), each level costs $\mathcal{O}(W)$ per window; Ridge inference is negligible compared to a single $f_\theta$ call. The total cost per forecast origin is $\mathcal{O}(|\mathcal{R}| \cdot C_\theta + |\mathcal{R}| \cdot W + (|\mathcal{R}|-1)C_{\text{ridge}})$, where $C_\theta$ is the cost of one call to $f_\theta$ and $C_{\text{ridge}}$ is the lightweight residual-prediction cost. The operator $\mathcal{P}_r$ is bounded because block averaging followed by repeat is an idempotent non-expansive projection, so $\|\mathcal{P}_r\|_2 \leq 1$ over the fixed resolution scale.

## C. Datasets, Regions, Covariates, and Splits

**Regions and time span.** We evaluate regional SPEI forecasting at three representative sites in South Australia. Table 7 lists coordinates, nominal climate zones, coverage, and sample counts. Coordinates refer to the center of the grid cell used to extract both covariates and the target series. The series is sampled at monthly temporal resolution from 1982/01 to 2018/12, yielding **444** months per site.

*Table 7.* Regions, climate zones, coverage, and sample counts at monthly resolution.

| Region ID | Coordinates (lat, lon) | Climate zone | Years covered | Total samples |
|---|---|---|---|---|
| SA-1 | $(-26.125, \ 129.125)$ | Arid | 1982 to 2018 | 444 |
| SA-2 | $(-29.125, \ 134.875)$ | Arid and seasonal | 1982 to 2018 | 444 |
| SA-3 | $(-35.625, \ 138.875)$ | Temperate | 1982 to 2018 | 444 |

**Target and data sources.** The forecasting target is **SPEI-30**, the Standardized Precipitation–Evapotranspiration Index computed using a 30-day accumulation window (Vicente-Serrano et al., 2010; Liu et al., 2024a). In this study, SPEI-30 is obtained from the global daily SPEI dataset (SPEI-GD) (Liu et al., 2024a), which provides daily SPEI at $0.25°$ spatial resolution for multiple accumulation timescales, including 5, 30, 90, 180, and 360 days. To align the daily SPEI-GD target with the monthly forecasting setting, we retain the last available daily SPEI-30 value in each calendar month; each monthly time stamp therefore represents the SPEI value accumulated over the preceding 30 days up to the end of that month. Positive values indicate wetter than normal conditions and negative values indicate drier than normal conditions. Target values are extracted at the study cell centers in Table 7. Meteorological covariates are taken from the *NCEP–NCAR Reanalysis 1* (Kalnay et al., 1996) as monthly fields on their native grids. These grids are variable-dependent: some variables are provided on regular $2.5° \times 2.5°$ latitude–longitude grids, while others are provided on T62 Gaussian grids with an approximate spatial resolution of $1.875°$. For each study location, NCEP–NCAR variables are sampled from the nearest native grid cell or bilinearly interpolated to the site coordinates as needed. Large-scale climate indices are obtained from public operational releases, including the Niño 1+2, Niño 3.4, and Niño 4 indices for ENSO (Trenberth, 1997), the Dipole Mode Index (DMI) for the Indian Ocean Dipole (Saji et al., 1999), and the Southern Annular Mode (SAM) index (Marshall, 2003). All sources are publicly available. The SPEI-GD product is accessible at `https://doi.org/10.5281/zenodo.8060268`. NCEP–NCAR Reanalysis 1 documentation and downloads are available from NOAA PSL at `https://www.psl.noaa.gov/data/gridded/data.ncep.reanalysis.html`. Public climate indices are available from operational centers such as NOAA CPC.

**Rolling origin evaluation and splits.** We use a rolling-origin evaluation with non-overlapping test origins. For each site, the 444 monthly observations from 1982/01 to 2018/12 are split chronologically into training, validation, and test periods

with a ratio of 70%/10%/20%, using split boundaries at the 70% and 80% points of each chronological series. Candidate residual predictors are fitted on the training split and selected by time-ordered validation; after $\lambda^{(k)}$ and $\delta^{(k)}$ are fixed, the final residual predictors are refit on the union of the training and validation splits and then frozen for test-time inference. The final test period is held out for reporting. The step between consecutive test origins equals the forecast horizon. The main text focuses on $H=1$, and Appendix G additionally reports $H \in \{1, 3, 6\}$ at Location 1. Per-channel standardization is fit on the training span only and applied unchanged to validation and test. No information from test targets is used in preprocessing, covariate construction, fitting, or selection. The resolution scale and all hyperparameters are held fixed across regions and splits with no retuning.

**Preprocessing and quality control.** Within channel missing values are linearly interpolated. Terminal gaps of at most two months are back filled or forward filled. After standardization, values beyond five median absolute deviations are clipped to the nearest bound. Calendar alignment ensures that the last index in each input window coincides with the forecast origin month.

**Evaluation protocol.** For each region and each horizon, we summarize the rolling-origin forecasts using MSE, MAE, and $R^2$ over the corresponding test span. The foundation baseline, the resolution scale, weighting rule, and step-size setting are identical to those used in the main text.

**Data availability and licensing.** All datasets used in this study are publicly available from the sources listed under "Target and data sources" above: SPEI-GD via Zenodo, NCEP–NCAR Reanalysis 1 via NOAA PSL, and the large-scale climate indices from operational centers. All series are obtained from public endpoints and do not require bespoke access.

**Computing environment and runtime.** Experiments are run on a single workstation equipped with an NVIDIA GeForce RTX 3090 graphics processor with 24 GB of memory, an Intel 12700K central processor, 64 GB of DDR4 3200 memory, and a 2 TB NVMe solid state drive. The operating system is Ubuntu 20.04.3 LTS. The software stack uses Python 3.10 with PyTorch 2.0.1 with CUDA 11.8, NumPy 1.24.3, Pandas 2.0.2, and Scikit learn 1.2.2. Random seeds are fixed for NumPy and PyTorch so that runs are deterministic where supported. The code repository includes configuration files and single command scripts that reproduce all scores and figures reported in the paper without retuning.

## D. Implementation Details and Hyperparameters

**Conventions (global).** All normalization statistics and residual predictors follow the leakage-safe protocol of App. C: statistics are computed on the training split only, hyperparameters (e.g., $\delta^{(k)}$ and $\lambda^{(k)}$) are selected on validation, and the predictors are then refit on training+validation and frozen before test evaluation.

### D.1. Projection operator, windowing, and boundary handling (main)

Unless marked otherwise, all experiments use the simple projection $\mathcal{P}_r = \mathcal{U}_r \circ \mathcal{D}_r$ exactly as defined in App. A.2 (block averaging, repeat upsampling, partial-block handling without future padding, and shared cross-channel timing), so $\mathcal{P}_r(\mathbf{X}_{t-W+1:t})$ always has length $W$.

*Context windowing.* At inference, the long context length is set to $W = \lfloor 0.7\,T \rfloor$ (where $T$ is the available series length at prediction time) so that the base TSFM receives a stable long-range context; the short window for learning residuals is defined in the original temporal resolution (Sec. 4), while the per-level stride $r_k$ only determines the effective number of downsampled observations, and is independent of $W$.

*Inference-time availability.* Under the rolling-origin evaluation protocol, the $p_{\text{lag}}$ residual lags in block (iv) are computed only from past revealed ground truth: once $y_t$ becomes observed at a later origin, we update the residual-history buffer with $e_t^{\text{hist}} = y_t - \widehat{y}_t^{(K)}$ for use by strictly later forecast origins (distinct from the analysis-only inference residual $\mathbf{R}^{(k)}$ in Sec. 4.5), so no target inside the current horizon is consumed by the current prediction.

The Ridge penalty $\lambda^{(k)}$ is selected by time-ordered validation on the log-grid $\{10^{-4}, 10^{-3}, \ldots, 10^2\}$.

### D.2. Residual-adaptive weighting and step size

**Residual-adaptive weighting.**   At level $k$, elementwise weights are

$$\boldsymbol{\omega}^{(k)} \;=\; \text{clip}_{[\varepsilon_{\min},1]}\Big(\sigma\big(\gamma\big(|\widehat{\mathbf{R}}^{(k)}| - \delta^{(k)}\big)\big)\Big), \qquad \sigma(x) = \tfrac{1}{1+e^{-x}},$$

where the logistic sharpness $\gamma{=}3.0$ is fixed (as in Sec. 4), $\delta^{(k)}$ is a *validation-selected* quantile of the absolute residual magnitude at level $k$ (quantile grid $\{0.60, 0.70, 0.75, 0.80, 0.85, 0.90\}$ unless stated), and $\varepsilon_{\min} > 0$ prevents zero weights.

**Step size.**   All refinement levels $k = 2, \ldots, K$ are executed, matching Algorithm 2. We use the fixed step size $\eta^{(k)} = 1.0$ in the reported experiments; the analysis permits any $\eta^{(k)} \in (0, 2]$ under the assumptions stated in Theorem 4.1. This setting is not treated as an independent ablation.

### D.3. Default hyperparameters and ranges

*Table 8.* Default hyperparameters and ranges (used unless stated otherwise).

| Parameter | Default | Range/Selection |
|---|---|---|
| Resolution scale $\mathcal{R}$ | $\{12, 6, 3, 2, 1\}$ | fixed |
| **Logistic sharpness (fixed)** | **3.0** | **fixed constant (matches main text)** |
| **Weight threshold $\delta^{(k)}$** | **val-quantile** | **grid $\{0.60, \ldots, 0.90\}$ (per level $k$)** |
| Weight floor $\varepsilon_{\min}$ | $10^{-3}$ | fixed |
| Step size $\eta^{(k)}$ | 1.0 | fixed in reported experiments; theory allows $(0, 2]$ |
| Mixing $\alpha^{(k)}$ | formula below | implied in $[0.3, 0.8]$ |
| Residual lags $p_{\text{lag}}$ | 6 | $\{3, 6\}$ |
| Trend window $q$ | 12 | $\{6, 12\}$ |
| Ridge penalty $\lambda^{(k)}$ | CV-selected | $\{10^{-4}, 10^{-3}, \ldots, 10^{2}\}$ (time-ordered CV) |

### D.4. Mixing coefficient (monotone coarse→fine)

The mixing coefficient increases with finer resolutions and satisfies $0 < \alpha^{(k)} < 1$:

$$\alpha^{(k)} \;=\; 0.3 \;+\; 0.5\left(1 - \frac{r_k}{\max(\mathcal{R})}\right),$$

where $r_k$ is the current stride and $\max(\mathcal{R}) = 12$ for $\mathcal{R} = \{12, 6, 3, 2, 1\}$, yielding $\alpha^{(k)} \in [0.3, 0.8]$. This choice aligns with the theory by placing more trust on finer-resolution proposals while retaining contraction.

### D.5. Remarks on hyperparameter economy

RGMR uses a small set of hyperparameters with default values (Tab. 8). When selection is required (e.g., $\delta^{(k)}$ quantiles, $\lambda^{(k)}$), we employ a *single* time-ordered validation pass. This keeps deployment overhead low while preserving the "plug-and-play" nature of the wrapper.

## E. Why South Australia and Societal Impact

**Rationale for choosing South Australia (SA).**   South Australia offers a stringent and policy relevant testbed for inference time adaptation because it concentrates diverse hydroclimate regimes within a compact domain. Coastal areas experience Mediterranean like seasonality and strong oceanic modulation while inland basins transition rapidly to semiarid and arid conditions, and this coastal to interior gradient, interacting with subseasonal bursts and multiyear swings driven by large scale modes such as ENSO, IOD, and SAM, produces multiscale variability that routinely exposes where single resolution forecasters underreact or overreact. For SPEI this appears as a standardized series with substantial mass near normal conditions and heavy tailed drought or wet anomalies, precisely the setting where our residual guided, multiresolution

refinement can correct coarse, temporally smeared signals without retraining the underlying foundation model. SA is also covered consistently by public gridded reanalysis at monthly resolution, so the same frozen base model and resolution scale can be evaluated across many locations under one rolling origin protocol without bespoke data access or regional retuning.

**Societal impact.** Sharper regional drought outlooks translate into concrete operational decisions: earlier and more reliable dry spell signals support water allocation, reservoir operations, and demand management; month ahead guidance for moisture deficits and heat stress improves agricultural scheduling and irrigation timing; and longer effective lead times help risk agencies and power systems preposition for bushfire seasons and demand peaks. These benefits matter most for remote and regional communities in arid zones that are disproportionately exposed to climate variability, where improving foresight at the horizons where operational choices are actually made enables targeted and equitable adaptation rather than reactive crisis management.

**Responsible use and limitations.** RGMR operates strictly at inference time on frozen foundation models. It cannot remove upstream data biases, coverage gaps, or structural errors in the base forecaster, and it should not be used as the sole basis for high stakes decisions. Best practice is to treat RGMR as one member in a multimodel and multievidence framework that includes expert judgment, observational context, and sector specific constraints. Reported improvements reflect the fixed resolution scale, weighting rule, and step-size settings specified in the main text. Changing those design choices or the evaluation geography may alter performance and uncertainty. To support scrutiny, extension, and safe reuse, we release the code with exact configurations and single command scripts mirroring the paper's rolling origin evaluation, so stakeholders can replicate results and reassess them in their own regional contexts before operational uptake.

# F. More Experimental Plots

**Setup.** We report additional one–month–ahead results at Location $(-26.125, 129.125)$ using the same leakage–safe protocol as the main paper (identical train/val/test splits, per–channel standardization fit on train, and $H=1$ unless stated). Figure 5 shows predicted vs. observed SPEI prediction plots for twelve baselines.

**Some findings.** (1) Foundation-scale models (e.g., TimesFM) tend to yield the tightest clouds, particularly near mild-to-moderate drought values. (2) Frequency/linear families (DLinear, FiLM) often reduce variance but can under-shoot extremes. (3) Patch- and pyramid-style Transformers (PatchTST, Pyraformer) better capture mesoscale variability yet may show scatter at the tails. (4) Nonstationarity-aware designs narrow bias under regime shifts but remain sensitive to rare events.

**Baselines at a glance.**

- **TimesFM** (Das et al., 2024). A decoder-only time series foundation model pretrained on large multi-domain corpora. It provides strong zero-shot and few-shot generalization for point forecasting with long context lengths, and it can be used as a frozen backbone with inference-time adaptation. Code: google-research/timesfm.

- **Transformer (vanilla)** (Vaswani et al., 2017). A capacity reference built from standard multi-head attention, positional encoding, and feed-forward blocks. Complexity scales quadratically with sequence length and it serves as a clean yardstick across datasets. Code: thuml/Time-Series-Library (model `Transformer.py` and unified training scripts).

- **Autoformer** (Wu et al., 2021). A decomposition-oriented Transformer that models trend and seasonality explicitly and replaces dot-product attention with series-wise autocorrelation to stabilize long-horizon prediction. Code: thuml/Autoformer.

- **Crossformer** (Zhang & Yan, 2023). A cross-dimension attention architecture for multivariate series. It introduces hierarchical embeddings and patching to capture inter-variable relations together with temporal dependencies at reduced cost. Code: Thinklab-SJTU/Crossformer.

- **DLinear** (Zeng et al., 2023). A minimal linear baseline that first separates trend and seasonal components by moving average then applies two one-layer linear heads and sums the results. Despite its simplicity it is competitive on long-horizon settings. Code: cure-lab/LTSF-Linear.

- **PatchTST** (Nie et al., 2023).    A patch-based Transformer that segments a long series into temporal patches as tokens and adopts channel-independent blocks so that each univariate channel shares weights. This design improves stability and efficiency for long horizons. Code: PatchTST/PatchTST.

- **TimeMixer** (Wang et al., 2024).    An MLP-style time-mixing model with multi-scale mixers that fuse short and long temporal patterns. It targets strong accuracy with training and inference efficiency and serves as a non-attention deep baseline. Code: kwuking/TimeMixer.

- **iTransformer** (Liu et al., 2024b).    An inverted formulation that treats time steps as tokens and embeds along the variable axis. This swap emphasizes temporal relations across long histories while keeping standard Transformer blocks. Code: thuml/iTransformer.

- **Pyraformer** (Liu et al., 2022a).    A pyramidal and dilated attention design that captures long-range context with sparse patterns and reduced attention cost. It is suited to very long contexts under limited compute. Code: ant-research/Pyraformer.

- **FiLM** (Zhou et al., 2022).    The frequency-improved Legendre memory model that uses Legendre state-space projection with frequency-domain enhancement and low-rank parameterization. It can act as a standalone predictor or a plug-in representation module. Code: DAMO-DI-ML/NeurIPS2022-FiLM.

- **MICN** (Wang et al., 2023).    A multi-scale model that decomposes series and applies inception-style convolutions to capture local details together with global context. It is a strong convolutional-style baseline for long-term forecasting. Code: wanghq21/MICN.

- **Nonstationary Transformer** (Liu et al., 2022b).    A framework that introduces series stationarization and de-stationarized attention to mitigate distribution shifts through time. It can be combined with several attention backbones including the vanilla Transformer. Code: thuml/Nonstationary_Transformers.

- **TimesNet** (Wu et al., 2023).    A 2D-variation modeling framework that reshapes 1D series into 2D tensors via FFT-discovered periods, capturing both intra- and inter-period patterns through Inception blocks. Code: thuml/Time-Series-Library.

- **TSMixer** (Chen et al., 2023).    An all-MLP architecture that mixes information along time and feature axes alternately, achieving strong forecasting accuracy with low parameter count and inference cost. Code: google-research/tsmixer.

**Unified scripts.**    For reviewers who prefer a single entry point, the THUML Time-Series-Library (Tsinghua Machine Learning Group, 2024) provides consistent data loaders, training scripts, and reference configurations that cover most baselines listed above.

# G. Extended Experiments and Clarifications

This appendix provides extended experiments including: (i) generalization across an additional variable (temperature) and three out-of-distribution regions; (ii) projection-operator ablation; (iii) clarification of methodological scope. These results complement the main text.

### G.1. More Generalization Across Variables and Regions

**Additional variable: Temperature.**    RGMR also improves temperature forecasting (Table 9), suggesting that its refinement is not SPEI-specific within the tested setting.

*Table 9.* Temperature forecasting results.

| Model | MSE↓ | MAE↓ | $R^2$↑ |
|---|---|---|---|
| TimesFM v1.0 | 3.121 | 1.534 | 0.933 |
| RGMR | **2.653** | **1.304** | **0.943** |

**Cross-region evaluation.**    We then test the generalization of RGMR by extending to three more regions outside South Australia (Table 10).

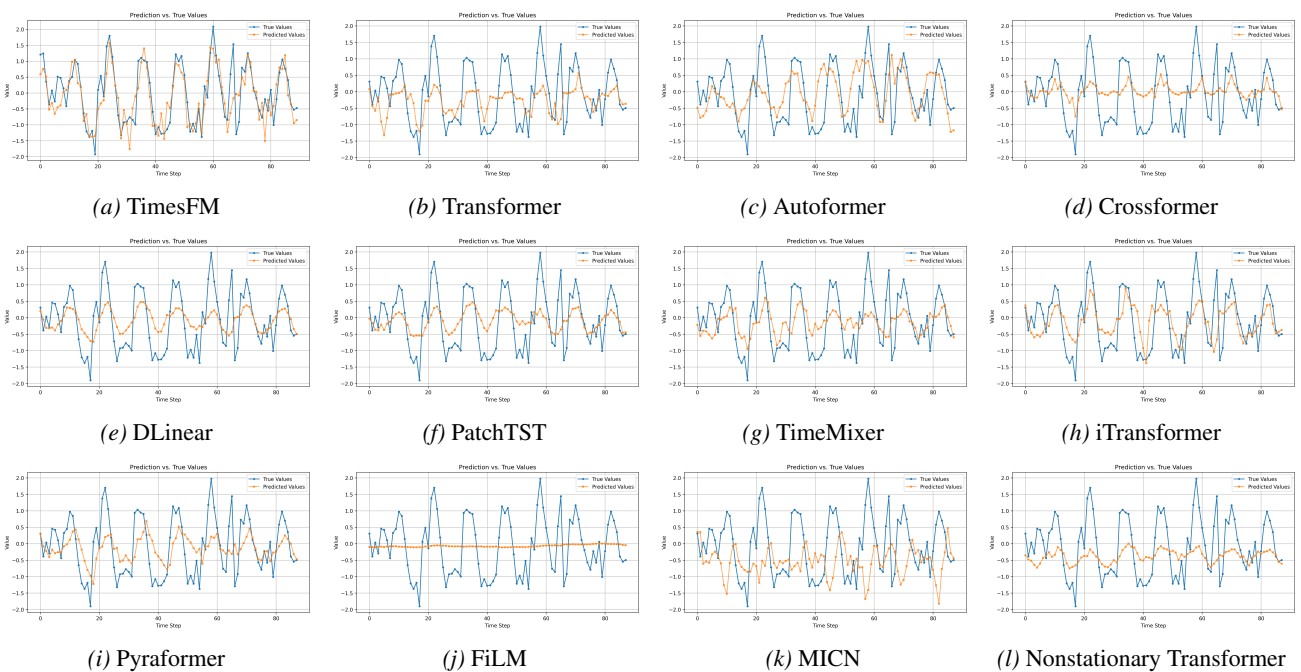

*(a)* TimesFM        *(b)* Transformer        *(c)* Autoformer        *(d)* Crossformer

*(e)* DLinear        *(f)* PatchTST        *(g)* TimeMixer        *(h)* iTransformer

*(i)* Pyraformer        *(j)* FiLM        *(k)* MICN        *(l)* Nonstationary Transformer

*Figure 5.* One–month–ahead SPEI at Location $(-26.125, 129.125)$ (appendix view). Each panel overlays *ground truth* (blue) and *model prediction* (orange) under the same leakage–safe protocol as the main paper (standardization fit on train; identical splits; $H=1$). TimesFM (foundation-scale) shows the tightest phase alignment on mild–moderate variability; linear/frequency families (e.g., DLinear, FiLM) reduce variance but may under-shoot extremes; patch/pyramid Transformers (PatchTST, Pyraformer) capture mesoscale fluctuations with occasional tail scatter; nonstationarity-aware variants alleviate regime-shift bias. No re-tuning is performed for this appendix figure.

*Table 10.* Cross-region generalization (TimesFM backbone).

| Region | Model | MSE↓ | MAE↓ | R$^2$↑ |
|---|---|---|---|---|
| US West Coast | TimesFM v1.0 | 0.155 | 0.313 | 0.854 |
| | RGMR | **0.081** | **0.222** | **0.924** |
| North Africa | TimesFM v1.0 | 0.101 | 0.261 | 0.892 |
| | RGMR | **0.065** | **0.201** | **0.933** |
| Indonesia | TimesFM v1.0 | 0.260 | 0.407 | 0.485 |
| | RGMR | **0.204** | **0.348** | **0.598** |

**Longer-horizon forecasting.** Because drought applications often require multi-month lead times, Table 11 reports results for $H = \{1, 3, 6\}$ months. RGMR improves the frozen TimesFM baseline across these tested horizons, indicating that the refinement mechanism extends beyond one-month-ahead forecasting in this setting.

*Table 11.* Forecasting performance at different horizons at Location 1 $(-26.125, 129.125)$. Lower is better for MSE/MAE; higher is better for $R^2$.

| Horizon | Model | MSE↓ | MAE↓ | R$^2$↑ |
|---|---|---|---|---|
| 1 | TimesFM v1.0 | 0.391 | 0.465 | 0.564 |
| | TimesFM v1.0 + RGMR | **0.318** | **0.423** | **0.651** |
| 3 | TimesFM v1.0 | 0.412 | 0.466 | 0.554 |
| | TimesFM v1.0 + RGMR | **0.345** | **0.433** | **0.627** |
| 6 | TimesFM v1.0 | 0.409 | 0.470 | 0.560 |
| | TimesFM v1.0 + RGMR | **0.376** | **0.465** | **0.606** |

### G.2. Projection Operator Ablation

We evaluated three projection operators used for downscaling to assess the projection choice: block averaging, wavelet decomposition, and STL decomposition. As shown in Table 12, simple block averaging provides the most stable and accurate refinement, whereas more complex decompositions tend to introduce unnecessary distortions at coarser scales.

*Table 12.* Projection operator comparison at Location 1 ($-26.125$, $129.125$).

| Projection | MSE↓ | MAE↓ | $R^2$↑ |
|---|---|---|---|
| Block Averaging | **0.318** | **0.423** | **0.651** |
| Wavelet | 0.366 | 0.467 | 0.597 |
| STL | 0.387 | 0.473 | 0.574 |

### G.3. Scope Clarification

RGMR is an inference-time refinement mechanism for frozen time-series foundation models. It does not modify model parameters; instead, it operates only on predictions and is compatible with any TSFM that exposes a standard forward interface, whether univariate or multivariate. This makes the refinement architecture-agnostic across the evaluated TSFM backbones and removes the need for retraining or task-specific parameter tuning.

We evaluate on SPEI rather than full climate fields: SPEI is a derived index aggregating multiple variables, and most operational systems model it directly. Full simulators such as SEAS5 (Johnson et al., 2019) and NMME (Kirtman et al., 2014) require high-dimensional atmospheric fields not used here and serve different scientific purposes, so a head-to-head comparison would not be meaningful.

RGMR also differs from test-time adaptation (Sun et al., 2020): TTA updates parameters online with auxiliary losses, while RGMR refines outputs without touching parameters. The two are therefore complementary and could in principle be stacked; we leave this combination to future work. Within the tested settings, our experiments show consistent gains across the regions, horizons, variables, backbones, resolution levels, and projection operators covered in this paper, with minimal overhead and behavior consistent with the contraction analysis.

