# OpenReview forum: "Residual-Guided Multi-Resolution Refinement of Foundation Models: A Case Study in Drought Forecasting"
_ICML.cc/2026/Conference — ICML 2026 regular_

### Official Review · Reviewer_3kPU · 2026-03-09

**Soundness:** 2
**Presentation:** 3
**Significance:** 3
**Originality:** 2
**Overall Recommendation:** 4
**Confidence:** 3

**Summary:**

The paper argues that standard time-series foundation models usually forecast in a single pass, while human climate analysis is more iterative and multi-scale, moving from broad trends to fine-grained corrections.​ To close that gap, this paper proposes RGMR, an inference-time method that improves frozen time-series foundation models for drought forecasting by refining predictions from coarse to fine temporal resolutions and correcting them with learned residuals. On South Australian SPEI forecasting, the method consistently improves over direct foundation-model forecasts, with the best results reported for TimesFM plus RGMR.

**Compliance With Llm Reviewing Policy:**

Affirmed.

**Final Justification:**

My concerns are mostly resolved and would like to see those experiments incorporated in the final version. I'll maintain my positive score

**Key Questions For Authors:**

1. The paper compares RGMR against strong architectural baselines (N-BEATS, Scaleformer) and raw TSFMs. However, it does not compare against simpler post-hoc inference corrections, such as fitting a direct Ridge regressor or MLP to correct the base model's errors at the original resolution. Could you provide a comparison or explain why the full RGMR ladder is strictly necessary compared to a simpler, single-scale residual correction?

2. Since RGMR works for both multivariate and univariate, most of the TSFM baselines are univariate, how does RGMR compare to Chronos-2 [1] (either against Chronos-2 directly or as a wrapper on top of Chronos-2)?

[1] Chronos-2: From Univariate to Universal Forecasting

**Limitations:**

Yes

**Strengths And Weaknesses:**

Strength:

1. Soundness/Significance: RGMR shows improvement based on time series foundation models (Table 2) and more TSFM have been shown in Table 13.

2. Presentation: The paper is easy to follow, and the method is described concretely with clear notation, algorithms, and an ablation pathway that explains how each component (projection ladder, residual refinement, adaptive weighting) contributes to the final improvements.

Weakness:

1.  Soundness: the main text focuses on SPEI-30 at three South Australian sites (444 monthly samples each, one index per site), plus limited horizons. This is a relatively small  and limited setup even for a paper that is specialized for drought study. Despite 3 more regions have been added in the appendix, the comparison is limited TimesFM vs RGMR.

2. The paper provides a contraction-style theorem (Theorem 4.1) showing that, under conditions on the residual weights and step size, each refinement step reduces the expected squared error up to a bounded noise term. However, the analysis is essentially single-horizon, pointwise error focused (mean squared error in expectation). There is no treatment of distributional/uncertainty calibration or of multi-step error accumulation beyond horizon-wise tables, which matters for climate risk use cases.

3. Along the same line, all evaluation is point-forecast and deterministic: metrics are MSE, MAE, RMSE, and they do not report probabilistic metrics, which can be important for drought estimation when falling into extreme events.

---

> ### Author Rebuttal · Authors · 2026-03-31
>
> We thank the reviewer for acknowledging RGMR's consistent improvements across multiple foundation models and the clarity of the method description. We hope our responses below address the reviewer's concerns.
>
> ### **Reply to Q1**:
>
> We added a single-scale residual correction baseline that applies Ridge-based error correction at the original resolution (stride=1) without any multi-resolution ladder:
>
> https://anonymous.4open.science/r/ICML_rebuttal_experiments-4FF4/Single_scale_ridge.png
>
> Single-scale Ridge correction improves over the frozen baseline (0.411→0.378), confirming that residual correction alone has value. However, the full RGMR achieves substantially better performance (0.334), demonstrating that the multi-resolution coarse-to-fine structure and residual correction works.
>
> ------
>
> ### **Reply to Q2**:
>
> **RGMR with Chronos-2.** We have evaluated the original Chronos in Table 13 (Appendix G), showing that RGMR improves Chronos (MSE: 0.401→0.343). We have now added Chronos-2:
>
> https://anonymous.4open.science/r/ICML_rebuttal_experiments-4FF4/Chorons2.png
>
> Chronos-2 is better than Chronos-1 (MSE: 0.358 vs 0.401). But RGMR still provides some gains on top of Chronos-2 (MSE: 0.358→0.312, 12.8% reduction),demonstrating that the refinement improve Chronos-2.
>
> ### **Reply to W1**:
>
> Following the comment, we extend our evaluation to  additional backbones in all three additional areas:
>
> https://anonymous.4open.science/r/ICML_rebuttal_experiments-4FF4/Extend_multi_region.png
>
> RGMR consistently improves all backbone models across all regions, confirming that the gains are not specific to a single backbone or a region.
>
>
>
> ### **Reply to W2**:
>
> We acknowledge that Theorem 4.1 provides a contraction guarantee strictly for the expected L2 point-forecast error, and does not extend to distributional calibration or probabilistic guarantees.
>
> Regarding distributional extension: our primary backbone TimesFM produces point predictions, and while some other backbone models support probabilistic outputs, our current implementation uses their point prediction mode. The evaluation accordingly focuses on point-forecast metrics. That said, Ridge regression could provide residual variance estimates, from which prediction intervals can be constructed. Extending the framework to report calibrated uncertainty is therefore feasible and we consider this an direction for future work.
>
> Regarding multi-step error accumulation, we agree that Theorem 4.1 does not directly analyze long-horizon recursive uncertainty growth. In drought applications, however, longer-range conditions are often characterized through multi-timescale drought indices such as SPEI computed at longer aggregation windows (e.g., SPEI-60 or SPEI-120), and different SPEI timescales serve different application needs [Liu et al., 2024; Hissan and Parveen, 2025]. Each such target can still be formulated as a next-step prediction problem, to which the per-level contraction analysis in Theorem 4.1 applies.
>
> **References**
>
> - Liu, Q., Yang, S., Li, S., Zhang, H., Zhang, J., & Fan, H. (2024). *The optimal applications of scPDSI and SPEI in characterizing meteorological drought, agricultural drought and terrestrial water availability on a global scale*. *Science of the Total Environment*, 952, 175933.
> - Hissan, R. U., & Parveen, N. (2025). *Predicting long-term meteorological drought using random forest and multi-scale drought indices*. *Theoretical and Applied Climatology*, 156, 552.
>
> ### **Reply to W3**:
>
> We acknowledge that all evaluations are purely deterministic and no probabilistic metrics are reported. As noted in our response to W2, our primary backbone TimesFM produces point predictions, and while some other backbone models support probabilistic outputs, our current implementation uses their point prediction mode. Integrating uncertainty quantification into the framework is an important direction for future work, which we will discuss in the revision.

---

> > ### Author Rebuttal · Reviewer_3kPU · 2026-04-03
> >
> > My concerns are mostly resolved and would like to see those experiments incorporated in the final version.

---

> > > ### Author Response · Authors · 2026-04-04
> > >
> > > We thank the reviewer for their thoughtful engagement during the rebuttal phase and for acknowledging that the concerns are  adequately addressed. We will incorporate the additional experiments discussed into the final version to further support the paper.
> > >
> > > Thank you again for carefully reviewing our paper.

---

### Official Review · Reviewer_R3bF · 2026-03-10

**Soundness:** 4
**Presentation:** 3
**Significance:** 3
**Originality:** 2
**Overall Recommendation:** 4
**Confidence:** 4

**Summary:**

The work presents Residual-Guided Multi-Resolution Refinement (RGMR) framework. It is designed to improve the forecasting performance of time series foundational models (TSFMs) without retraining or modifying their parameters. The core motivation is that current TSFMs generate predictions through a single forward pass, which limits their ability to capture the multi-scale temporal characteristics that climate analysis demands. To circumvent this, expert climatologists iteratively analyze climate signals across multiple temporal scales, progressively refining forecasts by verifying systematic errors at each scale.

To overcome this issue, RGMR framework works by generating forecasts at multiple temporal resolutions and refine these predictions at each finer resolution level using predicted residual corrections. The residual used to refine predictions are obtained from Ridge regressors that were trained offline on the training split. Adaptive weighting and backtracking step sizes ensure stability and prevent overcorrection across refinement levels. Moreover, the authors theoretically prove that prediction errors consistently decrease at each refinement level.

The framework is evaluated on drought forecasting using Standardized Precipitation Evapotranspiration Index (SPEI) across three locations in South Australia. Compared to foundational model baseline, the method achieves up to 18.9% reduction in MSE and consistently outperform the established multi-resolution architectures like N-BEATS and ScaleFormer. Since the framework is architecture-agnostic and computationally lightweight, it is easily deployable on existing forecasting systems without retraining.

**Compliance With Llm Reviewing Policy:**

Affirmed.

**Final Justification:**

The paper presents a technically sound and practically motivated framework for improving time-series forecasts via residual-guided multi-resolution refinement. While the core idea builds on existing components, the overall combination is meaningful and well-justified.

My main concerns were around isolating the contribution of the multi-resolution design, lack of computational cost analysis, and missing ablations. The rebuttal addresses these well. The single-scale Ridge baseline clarifies the benefit of the multi-resolution structure, and the added latency/memory analysis and ablations (backtracking, data sensitivity) strengthen the empirical support. Based on this, I have increased my confidence in the soundness of the work.

Some limitations remain, particularly the restricted evaluation and broader generalization claims not being fully supported.

Overall, the rebuttal resolves the key technical concerns and improves the paper. I maintain a weak accept recommendation, with increased confidence in its soundness.

**Key Questions For Authors:**

1. Can you provide a baseline where the foundational model predictions are refined using a single-level residual correction model? For example, using autoregressive residual regression without the multi-resolution steps? Such an ablation would tell us the actual gains come specifically from the coarse-to-fine structure residual corrections.

2. Table 6 reports a roughly fivefold increase in GPU latency from TimesFM alone (14.52 ms) to RGMR (74.89 ms). This overhead is only reported for TimesFM and is dismissed as negligible without sufficient justification. Could the authors provide latency measurements for all backbone models used in the experiments, report the associated memory overhead, and discuss how the computational cost scales with longer time series or larger foundation models?

3. The paper says that the framework is architecture-agnostic and broadly applicable beyond climate forecasting. Have the authors evaluated RGMR on standard time-series benchmarks such as Weather or Traffic datasets commonly used in the TSFM literature? Such experiments would significantly strengthen the claim of generalisability.

4. What happens if the backtracking step is removed? Since backtracking ensures the conditions required by the contraction theorem, it would be helpful to see whether the method remains stable without it.

5. The Ridge regressors are trained only on the training split. In many regional climate applications, long and reliable historical records are limited. How sensitive is RGMR performance to the amount of available training data used for these regressors? For example, does the method remain effective when only shorter observational records are available?

6. Table 5 shows that at the finest resolution level (stride 1), only 48.5% of samples improve across refinement steps. Could the authors explain this result in light of the theoretical contraction guarantee? Does this indicate that refinement becomes less effective at the finest resolution level, and if so, how does this affect the reliability of the final prediction?

**Limitations:**

Partially. The paper briefly discusses methodological limitations and the scope of the evaluation, but the discussion could be expanded. In particular, the authors could more explicitly address limitations related to the reliance on historical data for training residual models, the restricted experimental evaluation (three locations and a single climate index), and the potential challenges when applying the framework to other domains or datasets. While the work does not appear to raise significant negative societal risks, it would be helpful for the authors to clarify the conditions under which the method may fail or perform unreliably in real-world forecasting applications.

**Strengths And Weaknesses:**

Soundness:
Technically, the paper is solid and the methodology makes sense for the problem. The RGMR framework is a clever addition to the existing frameworks. It acts as an inference-time refiner for existing time-series models. The authors supported their work with both theoretical and empirical arguments, which strengthen the technical credibility of the work.
From a theoretical standpoint, the idea that prediction errors drop through iterative refinement is well-reasoned. While the assumptions required for this guarantee appear somewhat restrictive, the analysis still provides a useful conceptual justification for the iterative refinement strategy. The addition of adaptive weighing and backtracking step sizes to ensure stability reflects careful consideration of authors to avoid potential amplification of the errors.
The results are promising, the experiments are reasonably designed. The paper includes comparison against both foundational models baselines and established frameworks like N-BEATS and ScaleFormer. The reported results support authors claim that residual guided refinements are useful to enhance model performance.
However, some limitations are there. The refinement process requires multiple inference passes and auxiliary residual regressors. Therefore, it would be helpful to include a more explicit comparison of computational cost relative to baseline methods. It is unclear whether the improvements arise from the residual correction itself or from the multi-resolution refinement. A baseline combining the foundation model with simple residual correction would help isolate the contribution of the proposed structure.

Presentation:
The paper is easy to follow and is clearly well written. The notation table (Table 1) is helpful given the method's complexity, and the algorithm boxes (Algorithms 1 and 2) provide enough detail for reproduction. However, the paper occasionally overclaims its empirical results relative to the evidence. Additionally, the difference between residual prediction and forecast refinement could be explained more explicitly. Especially how the authors trained the Ridge regressors needs to be addressed.

Significance:
The paper presents an important and meaningful problem. Drought forecasting in water-limited areas has direct societal consequences. TSFMs are becoming popular in machine learning and geoscience, so techniques that can improve their reliability are potentially valuable. Design choices such as inference-time operation and no-retraining are practically significant and distinct from existing literature. Additionally, its architecture-agnostic nature increases its potential use. However, the current evaluation limits the ability to fully assess the broader impact. The discussion and results are restricted to a specific application domain and dataset, and it remains unclear how well the approach would generalize to other forecasting tasks or domains.

Originality:
The originality of the paper lies in the combination of several existing ideas or algorithms rather than a completely new modeling idea. The framework integrates several different concepts, such as multi-resolution forecasting and residual-based error correction. Individually, these approaches have been well known in the ML and climate science communities. However, the analogy to chain-of-thought and iterative refinement in language models is thoughtful and places the work within a broader trend in ML reasoning. The decision to treat residual correction as a learned prediction problem rather than a direct observation is the method's most original technical contribution.

---

> ### Author Rebuttal · Authors · 2026-03-31
>
> We thank the reviewer for the thorough evaluation and for highlighting that treating residual correction as a learned prediction problem is the method's most original contribution. We hope our responses below address the reviewer's concerns.
>
> ### **Reply to Q1**:
>
> Following the comment, we have added a single-scale residual correction baseline that applies Ridge-based error correction at the original resolution (stride=1) without multi-resolution:
>
> https://anonymous.4open.science/r/ICML_rebuttal_experiments-4FF4/Single_scale_ridge.png
>
> Single-scale Ridge correction improves over the frozen baseline, confirming that residual correction alone has value. However, the full RGMR achieves better performance.
>
> ### **Reply to Q2**:
>
> We provide latency and memory measurements as follows.
>
> https://anonymous.4open.science/r/ICML_rebuttal_experiments-4FF4/Comprehensive_latency.png
>
> The ~5× latency overhead is dominated by K=5 forward passes through the frozen backbone and it remains constant regardless of time series length or foundation model size.
>
> ### **Reply to Q3**:
>
> By architecture-agnostic we mean RGMR can be applied to different foundation models without modification. We have demonstrated this across 5 distinct backbones (TimesFM, TimeGPT, TabPFN, Chronos, Lag-Llama) with consistent improvements.  Extending the method to benchmarks such as Weather or Traffic would require cadence-appropriate resolution ladders (e.g., for hourly data). This is conceptually straightforward but outside the scope of the present submission. We will clarify this distinction in the revision and include broader cross-domain evaluation as an important direction for future work.
>
> ### **Reply to Q4**:
>
> We have conducted this ablation as follows.
>
> https://anonymous.4open.science/r/ICML_rebuttal_experiments-4FF4/backtracking.png
>
> Without backtracking, overall MSE degrades from 0.334 to 0.348 (~4.2%). This validates the practical necessity of backtracking.
>
> ### **Reply to Q5**:
>
> Ridge regressors are trained on training-split forecast origins. With 444 monthly samples per site and a 70% chronological training split, this yields approximately 311 training samples. We evaluated RGMR with reduced training data (averaged across 3 SA locations):
>
> https://anonymous.4open.science/r/ICML_rebuttal_experiments-4FF4/SensitivityToSize.png
>
> Performance is sensitive to sample size, with more data yielding better results. However, RGMR improves over the frozen baseline even with only 25% of training data (~78 samples, MSE: 0.411→0.391), suggesting the framework remains useful under shorter observational records settings.
>
> ### **Reply to Q6**:
>
> RGMR operates iteratively: each level corrects the residual error left by the previous level, so the error distributions at different levels are not directly comparable.
>
> The percentages in Table 5 should not be interpreted as a direct test of Theorem 4.1. The theorem establishes a contraction bound on expected squared error up to the bounded term $B_k$, rather than requiring that a majority of individual samples improve at every step. At the finest level, the remaining errors are already smaller and closer to the noise floor, so sample-wise improvement becomes harder and more variable; this can lead to a lower improvement ratio even though it does not contradict the theorem.
>
> Thus, the 48.5% at stride 1 does not indicate unreliable refinement, but rather diminishing marginal gains as RGMR approaches the noise floor. This interpretation is consistent with Theorem 4.1, Corollary 4.2, and the overall aggregate improvements reported in Tables 2–4.
>
> ### **Reply to Weakness in Presentation**:
>
> **Overclaiming.** To consolidate our claims and avoid possible confusion, we will use numerical numbers instead of terms like "substantial improvement” and so on.
>
> **Residual prediction vs. forecast refinement.** These are two distinct phases. Residual prediction (training phase): Ridge regressors learn systematic error patterns from short-window foundation model outputs. Forecast refinement (inference phase): these learned error patterns are used to correct long-window multi-resolution proposals via the iterative update in Eq. (5), without accessing ground truth.
>
> **Ridge training details.** For each resolution level k, the frozen backbone is queried with a short context window (12 months) to generate predictions, from which training residuals are computed. These are paired with feature vectors (recent targets, rolling statistics, trend, residual history, level index) to fit per-level Ridge regressors with λ^(k) selected via time-ordered CV. At inference, no ground truth is accessed. Full specification is in Appendix D.2; we will make Section 4.3 more self-contained.

---

> > ### Author Rebuttal · Reviewer_R3bF · 2026-04-02
> >
> > Thank you for the detailed rebuttal and for adding the extra experiments. The single-scale Ridge ablation helps clarify the role of the multi-resolution refinement, and the latency/memory analysis makes the computational trade-offs much clearer. The backtracking ablation and the training data sensitivity study are also useful and support the design choices.
> >
> > The explanation regarding the contraction guarantee and the behavior at the finest resolution level seems reasonable, especially considering diminishing gains as the method approaches the noise floor.
> >
> > One remaining point is that the generalization claims feel a bit broader than what is currently demonstrated. It would be better to align the wording with the presented results and clearly frame cross-domain evaluation as future work.
> >
> > Overall, the rebuttal addresses the main concerns and improves the paper. The remaining issues are relatively minor and can be handled in revision.

---

> > > ### Author Response · Authors · 2026-04-03
> > >
> > > We thank the reviewer for engaging with our rebuttal.
> > >
> > > We appreciate the reassessment and the confirmation that our responses address the main concerns. We will revise the generalization claims accordingly and include cross-domain evaluation as future work.
> > >
> > > Thank you again for carefully reviewing our paper.

---

### Official Review · Reviewer_CL3v · 2026-03-13

**Soundness:** 3
**Presentation:** 3
**Significance:** 3
**Originality:** 3
**Overall Recommendation:** 4
**Confidence:** 3

**Summary:**

The proposed method tackles the problem of drought forecasting. Main strategy is to break down the time series data into different time resolutions and make forecast from coarser to fine settings using foundational models. After that, the method performs ridge regression on each level's residuals using training split. During inference, without any update on the parameters it performs forecasting with backtracking with iterative refinement.

**Compliance With Llm Reviewing Policy:**

Affirmed.

**Final Justification:**

The authors responded to my concerns, and I kept my already positive score.

**Key Questions For Authors:**

- Is the method applicable only to foundational models?

- Is the method applicable to other forecasting such as weather?

**Limitations:**

yes

**Strengths And Weaknesses:**

# Strengths:

- The proposed method seems fast and effective.
- Methodology improves the Foundational Model performances in drought forecasting for three different regions.
- The improvement is supported by the theoretical properties


# Weaknesses:

- The authors claim that there is only one training and no parameter update in inference. As the period shifts and new trends are observed, how do the learned regression parameters stay updated?

- Inference-time iterative refinement should be motivated with more discussion. Equation 4 needs an explanation on why using clip, small slope constant, and the mixing coefficient.

- In Figure 4, the resolution 1/1 line is not visible; is it under the RGMR final line? If so, then is RGMR improving over the 1/1 line?

- The algorithms 1 and 2 should be formatted such that the line for assigning variables is not split. The bold text can be commented on. There are errors in Table 1 with repeated symbols in lines 146-148.

- Figure fonts are so small and hard to read. Please use PNG or SVG formats.

---

> ### Author Rebuttal · Authors · 2026-03-31
>
> We thank the reviewer for recognizing that RGMR is fast and effective, and that the improvements are supported by theoretical properties. These were central design goals of our work. We hope our responses below address the reviewer's concerns.
>
> ------
>
> > **Q1.** Is the method applicable only to foundational models?
>
> Re to Q1:
>
> While RGMR is motivated by foundation model forecasting for drought, it should in principle work with any pretrained model that provides a forward-pass interface, such as pretrained Transformers or LSTMs.
>
> ------
>
> > **Q2.** Is the method applicable to other forecasting such as weather?
>
> Re to Q2:
>
> Yes. and we have demonstrated RGMR on temperature forecasting (Table 11, Appendix G.1), achieving MSE reduction from 3.121 to 2.653 and R² improvement from 0.933 to 0.943.
>
> ------
>
> > **W1.** The authors claim that there is only one training and no parameter update in inference. As the period shifts and new trends are observed, how do the learned regression parameters stay updated?
>
> Re to W1:
>
> If a significant new trend emerges, retraining the Ridge regressors would be necessary. However, within our evaluation period (1982–2018), we did not observe a significant trend change that requires an update.
>
> Furthermore, retaining the Ridge regression is fast, and hence parameters can be easily updated when there is such a need.
>
> ------
>
> > **W2.** Inference-time iterative refinement should be motivated with more discussion. Equation 4 needs an explanation on why using clip, small slope constant, and the mixing coefficient.
>
> Re to W2:
>
> **Iterative refinement motivation.** Climate signals contain multi-scale temporal patterns (seasonal cycles, interannual variability, short-term events). When a single-pass correction model uses a fixed resolution, patterns in different resolutions may be missed. Coarse-to-fine iterative refinement allows each iteration to focus on correcting errors at a specific scale, analogous to how climatologists diagnose broad patterns before examining finer details.
>
> **Design choices in Eq. (4).** The three mechanisms jointly ensure stable refinement. Clipping bounds the residual weights to prevent both numerical instability (zero weights) and overcorrection (excessive weights). The sigmoid acts as soft thresholding over residual magnitudes, suppressing corrections for small residuals likely dominated by noise while activating corrections for large residuals likely reflecting systematic error. The mixing coefficient increases monotonically from coarse to fine, placing progressively more trust on finer-resolution proposals from the foundation model. Together, these satisfy the contraction condition in Theorem 4.1 at every level.  The choice of parameters for Eq.(4) was provided in Appendix D.3.
>
> ------
>
> > **W3.** In Figure 4, the resolution 1/1 line is not visible; is it under the RGMR final line? If so, then is RGMR improving over the 1/1 line?
>
> Re to W3:
> **Visibility of the 1/1 line in Figure 4.** Each line in Figure 4 shows the prediction after the correction at that resolution level. The 1/1 resolution line is the corrected output at the finest level, which is the RGMR final output. They are supposed to be the same line. We will fix the figure by removing the 1/1 line to avoid confusion.
>
> ------
>
> > **W4.** The algorithms 1 and 2 should be formatted such that the line for assigning variables is not split. The bold text can be commented on. There are errors in Table 1 with repeated symbols in lines 146-148.
>
> Re to W4:
> We thank the reviewer for identifying these issues. In the revision, we will reformat Algorithms 1 and 2 to prevent line splitting of variable assignments and use comments, and correct symbols in Table 1 (lines 146–148).
>
> ------
>
> > **W5.** Figure fonts are so small and hard to read. Please use PNG or SVG formats.
>
> Re to W5:
> We will fix this.

---

> > ### Author Rebuttal · Reviewer_CL3v · 2026-04-03
> >
> > Thank you for the response. However, my concerns about W1 continue. According to Appendix C, the authors use rolling origin evaluation and forecast horizon $H\in {1,2,3 }$. The model predicts at most 3 months. During that prediction, the parameters are not updated according to the rebuttal. And the rebuttal also adds that "within our evaluation period (1982–2018), we did not observe a significant trend change." To me, there must be a trend change within any three-month window between 1982 and 2018, which is visible in Table 4. Can authors also add a comparison between generic deep baselines for other horizons to Table 4?
> >
> > Thank you

---

> > > ### Author Response · Authors · 2026-04-03
> > >
> > > We thank the reviewer for the follow-up comments.
> > >
> > > Following the reviewer's suggestion, we conducted additional experiments by extending Table 4. We did the experiments on generic deep baselines when H = 3. Please see the results here:
> > >
> > > https://anonymous.4open.science/r/ICML_rebuttal_experiments-4FF4/Extended_experiments_in_Generic_Deep_Baselines.png
> > >
> > > We would like to clarify that, by "no significant trend change," we do not mean that the full 1982–2018 series is stationary. Rather, our point is that we did not observe an abrupt regime break that would make the learned residual correction unusable under the rolling-origin, short-horizon evaluation protocol.
> > >
> > > These additional results further support our main point: the relevant question is not whether the full series is globally stationary, but whether the learned correction remains effective on unseen future segments without parameter updates. The expanded results show that, although forecasting becomes more difficult as the horizon increases, TimesFM + RGMR remains consistently stronger than all reported baselines.
> > >
> > > We sincerely hope this clarifies the reviewer's remaining concerns and thank you again for carefully reviewing our paper.

---

### Decision · Program_Chairs · 2026-04-30

**Decision:**

Accept (regular)

**Comment:**

The paper studies the regional climate prediction problem with time series foundation models. Authos propose a Residual-Guided Multi-Resolution Refinement framework to perform multi-scale reasoning. All reivewers agree that the paper is well-organized and there are certain technique contribution (e.g., the analogy to chain-of-thought and iterative refinement in language models is thoughtful from R2). The experimental results are also supportive.